# On the $\mathcal{O}(\frac{\sqrt{d}}{K^{1/4}})$ Convergence Rate of AdamW Measured by $\ell_1$ Norm

**Huan Li** [1] ✉
lihuanss@nankai.edu.cn

**Yiming Dong** [2]
ymdong@stu.pku.edu.cn

**Zhouchen Lin** [2,3,4] ✉
zlin@pku.edu.cn

1. College of Artificial Intelligence, Nankai University, Tianjin, China
2. State Key Lab of General AI, School of Intelligence Science and Technology, Peking University
3. Institute for Artificial Intelligence, Peking University, Beijing, China
4. Pazhou Laboratory (Huangpu), Guangzhou, China

## Abstract

As the default optimizer for training large language models, AdamW has achieved remarkable success in deep learning. However, its convergence behavior is not theoretically well-understood. This paper establishes the convergence rate $\frac{1}{K}\sum_{k=1}^{K}\mathbb{E}\left[\|\nabla f(\mathbf{x}^k)\|_1\right] \leq \mathcal{O}(\frac{\sqrt{d}C}{K^{1/4}})$ for AdamW measured by $\ell_1$ norm, where $K$ represents the iteration number, $d$ denotes the model dimension, and $C$ matches the constant in the optimal convergence rate of SGD. Theoretically, we have $\|\nabla f(\mathbf{x})\|_2 \ll \|\nabla f(\mathbf{x})\|_1 \leq \sqrt{d}\|\nabla f(\mathbf{x})\|_2$ for any high-dimensional vector $\mathbf{x}$ and $\mathbb{E}\left[\|\nabla f(\mathbf{x})\|_1\right] \geq \sqrt{\frac{2d}{\pi}}\mathbb{E}\left[\|\nabla f(\mathbf{x})\|_2\right]$ when each element of $\nabla f(\mathbf{x})$ is generated from Gaussian distribution $\mathcal{N}(0,1)$. Empirically, our experimental results on real-world deep learning tasks reveal $\|\nabla f(\mathbf{x})\|_1 = \Theta(\sqrt{d})\|\nabla f(\mathbf{x})\|_2$. Both support that our convergence rate can be considered to be analogous to the optimal $\frac{1}{K}\sum_{k=1}^{K}\mathbb{E}\left[\|\nabla f(\mathbf{x}^k)\|_2\right] \leq \mathcal{O}(\frac{C}{K^{1/4}})$ convergence rate of SGD.

## 1 Introduction

AdamW, which modifies Adam by decoupling weight decay from gradient-based updates, has emerged as the dominant optimizer for training deep neural networks, particularly for large language models. AdamW represents the pinnacle of adaptive gradient algorithms, having developed through the progression of AdaGrad [1, 2], RMSProp [3], Adam [4], and finally AdamW [5] itself. Although the literature on the convergence analysis of adaptive gradient algorithms is quite extensive, there has been little research on the convergence properties of AdamW.

Recently, Xie and Li [6] proved that if the iterates of AdamW converge to some $\mathbf{x}_\infty$, then $\mathbf{x}_\infty$ is a KKT point of the constrained problem

$$\min_{\mathbf{x}\in\mathbb{R}^d} f(\mathbf{x}), \quad s.t. \quad \|\mathbf{x}\|_\infty \leq \frac{1}{\lambda}, \tag{1}$$

where $f(\mathbf{x})$ is the nonconvex objective function, $\|\cdot\|_\infty$ is the infinity norm, and $\lambda$ is the weight decay parameter. Moreover, $\mathbf{x}$ is a KKT point of problem (1) iff [6]

$$\|\mathbf{x}\|_\infty \leq \frac{1}{\lambda} \quad \text{and} \quad \langle\lambda\mathbf{x}, \nabla f(\mathbf{x})\rangle + \|\nabla f(\mathbf{x})\|_1 = 0. \tag{2}$$

Xie and Li characterized which solution does AdamW converge to, if it indeed converges. The next fundamental question to address is whether and how fast AdamW converges. Zhou et al. [7]

39th Conference on Neural Information Processing Systems (NeurIPS 2025).

conducted preliminary exploration on this problem. However, their analysis requires the weight decay parameter to decrease exponentially, making AdamW reduce to Adam finally. To the best of our knowledge, aside from [7], we have not found any other literature addressing the convergence issue of AdamW.

In practical deep learning training, we often initialize the network weights small and employ modest weight decay, for example, $\lambda = 0.01$, which empirically confines the optimization trajectory within the $\ell_\infty$ norm constraint, as empirically demonstrated in Figure 3. That is, $\|\mathbf{x}\|_\infty \leq \frac{c}{\lambda}$ for some $c < 1$, making $\langle \lambda \mathbf{x}, \nabla f(\mathbf{x}) \rangle + \|\nabla f(\mathbf{x})\|_1$ lower bounded by $(1-c)\|\nabla f(\mathbf{x})\|_1$. This key property enables the use of $\|\nabla f(\mathbf{x})\|_1$ as an effective yet significantly simpler convergence metric for AdamW in practical settings.

Building on the above observation, this paper focuses on the convergence rate of AdamW within the constraint in problem (1). Specifically, we prove the following convergence rate for AdamW

$$\frac{1}{K} \sum_{k=1}^{K} \mathbb{E}\left[\|\nabla f(\mathbf{x}^k)\|_1\right] \leq \mathcal{O}\left(\frac{\sqrt{d}}{K^{1/4}} \sqrt[4]{\sigma_s^2 L(f(\mathbf{x}^1) - f^*)} + \sqrt{\frac{dL(f(\mathbf{x}^1) - f^*)}{K}}\right) \tag{3}$$

by proper parameter settings such that $\|\mathbf{x}^k\|_\infty < \frac{1}{\lambda}$ for all iterates, where $K$ is the total iteration number, $d$ is the model dimension, $\sigma_s$ is the gradient noise variance, $L$ is the Lipschtiz smooth constant, and $f^*$ is a lower bound of $f(\mathbf{x})$. Recall the classical convergence rate of SGD [8]

$$\frac{1}{K} \sum_{k=1}^{K} \mathbb{E}\left[\|\nabla f(\mathbf{x}^k)\|_2\right] \leq \mathcal{O}\left(\frac{\sqrt[4]{\sigma_s^2 L(f(\mathbf{x}^1) - f^*)}}{K^{1/4}}\right), \tag{4}$$

which matches the lower bound of nonconvex stochastic optimization [9]. Comparing (3) with (4), we see that our convergence rate (3) also achieves the same lower bound with respect to $K$, $\sigma_s$, $L$, and $f(\mathbf{x}^1) - f^*$. The only coefficient left unclear whether it is tight is the dimension $d$. Theoretically, we have $\|\nabla f(\mathbf{x})\|_2 \ll \|\nabla f(\mathbf{x})\|_1 \leq \sqrt{d}\|\nabla f(\mathbf{x})\|_2$ for any high-dimensional vector $\mathbf{x}$ and $\mathbb{E}\left[\|\nabla f(\mathbf{x})\|_1\right] \geq \sqrt{\frac{2d}{\pi}}\mathbb{E}\left[\|\nabla f(\mathbf{x})\|_2\right]$ when each element of $\nabla f(\mathbf{x})$ is generated from Gaussian distribution $\mathcal{N}(0,1)$. Empirically, we have observed $\|\nabla f(\mathbf{x})\|_1 = \Theta(\sqrt{d})\|\nabla f(\mathbf{x})\|_2$ on real-world deep learning tasks, as shown in Figure 2. Thus, we could say that our convergence rate (3) can be considered to be analogous to (4) of SGD in the ideal case.

As a special case, we also establish the same convergence rate (3) for Adam under slightly relaxed parameter settings than AdamW. To the best of our knowledge, this convergence rate only appears for RMSProp firstly proved in [10], and similar results for AdaGrad subsequently appeared in [11, 12] and RMSProp in [13] under different assumptions. Notably, comparable convergence guarantees remain unproven for AdamW and Adam.

## 2 Convergence Rate of AdamW

This section presents our convergence rate analysis for AdamW. We first describe the assumptions used throughout this paper as follows, where we denote $\mathcal{F}_k = \sigma(\mathbf{g}^1, \mathbf{g}^2, \cdots, \mathbf{g}^k)$ to be the sigma field of the stochastic gradients up to $k$, denote $\mathbb{E}_{\mathcal{F}_k}[\cdot]$ as the expectation with respect to $\mathcal{F}_k$ and $\mathbb{E}_k[\cdot|\mathcal{F}_{k-1}]$ the conditional expectation with respect to $\mathbf{g}^k$ conditioned on $\mathcal{F}_{k-1}$.

Assumptions:

1. Smoothness:
   $\|\nabla f(\mathbf{y}) - \nabla f(\mathbf{x})\| \leq L\|\mathbf{y} - \mathbf{x}\|$,

2. Unbiased estimator:
   $\mathbb{E}_k\left[\mathbf{g}^k|\mathcal{F}_{k-1}\right] = \nabla f(\mathbf{x}^k)$,

3. Coordinate-wise bounded noise variance:
   $\mathbb{E}_k\left[|\mathbf{g}_i^k - \nabla_i f(\mathbf{x}^k)|^2|\mathcal{F}_{k-1}\right] \leq \sigma_i^2$.

---

**Algorithm 1** AdamW

Hyper parameters: $\eta, \theta, \beta, \lambda, \varepsilon$
Initialize $\mathbf{x}^1$, $\mathbf{m}^0 = 0$, $\mathbf{v}^0 = 0$
**for** $k = 1, 2, \cdots, K$ **do**
    $\mathbf{g}^k = \text{GradOracle}(\mathbf{x}^k)$
    $\mathbf{m}^k = \theta\mathbf{m}^{k-1} + (1-\theta)\mathbf{g}^k$
    $\mathbf{v}^k = \beta\mathbf{v}^{k-1} + (1-\beta)(\mathbf{g}^k)^{\odot 2}$
    $\mathbf{x}^{k+1} = (1-\lambda\eta)\mathbf{x}^k - \frac{\eta}{\sqrt{\mathbf{v}^k}+\varepsilon} \odot \mathbf{m}^k$
**end for**

---

Denoting $\boldsymbol{\sigma} = [\sigma_1, \cdots, \sigma_d]$ as the noise variance vector and $\sigma_s = \|\boldsymbol{\sigma}\|_2 = \sqrt{\sum_{i=1}^{d} \sigma_i^2}$, we have the

following standard bounded noise variance assumption

$$\mathbb{E}_k\left[\|\mathbf{g}^k - \nabla f(\mathbf{x}^k)\|^2 \big| \mathcal{F}_{k-1}\right] \le \sigma_s^2.$$

Algorithm 1 provides the complete AdamW implementation, where we denote $\odot$ for the Hadamard product. Setting the weight decay parameter $\lambda = 0$ recovers the standard Adam. For analytical simplicity, we omit the bias correction term in our analysis.

Based on Assumptions 1-3, we provide the convergence rate of AdamW in the following theorem. Note that we do not assume the boundedness of the gradient $\nabla f(\mathbf{x}^k)$ or stochastic gradient $\mathbf{g}^k$.

**Theorem 1** *Suppose that Assumptions 1-3 hold. Define $\hat{\sigma}_s^2 = \max\left\{\sigma_s^2, \frac{L(f(\mathbf{x}^1)-f^*)}{K\gamma^2}\right\}$ with any constant $\gamma \in (0,1]$. Let $1 - \theta = \sqrt{\frac{L(f(\mathbf{x}^1)-f^*)}{K\hat{\sigma}_s^2}}$, $\theta \le \beta \le \sqrt{\theta}$[1], $\eta = \sqrt{\frac{f(\mathbf{x}^1)-f^*}{4KdL}}$, $\varepsilon = \frac{\hat{\sigma}_s^2}{d}$, $\lambda \le \frac{\sqrt{d}}{\sqrt{72}K^{3/4}}\sqrt[4]{\frac{L^3}{\hat{\sigma}_s^2(f(\mathbf{x}^1)-f^*)}}$, and $\|\mathbf{x}^1\|_\infty \le \sqrt{\frac{K(f(\mathbf{x}^1)-f^*)}{dL}}$. Then for AdamW, we have $\|\mathbf{x}^k\|_\infty < \frac{1}{\lambda}$ for all $k = 1, 2, \cdots, K$ and*

$$\frac{1}{K}\sum_{k=1}^K \mathbb{E}\left[\|\nabla f(\mathbf{x}^k)\|_1\right] \le \frac{8\sqrt{d}}{K^{1/4}}\sqrt[4]{\hat{\sigma}_s^2 L(f(\mathbf{x}^1)-f^*)} + 30\sqrt{\frac{dL(f(\mathbf{x}^1)-f^*)}{K}}.$$

*Specially, when $\sigma_s^2 \le \frac{L(f(\mathbf{x}^1)-f^*)}{K\gamma^2}$, we have $1 - \theta = \gamma$, $\theta \le \beta \le \sqrt{\theta}$, $\eta = \sqrt{\frac{f(\mathbf{x}^1)-f^*}{4KdL}}$, $\varepsilon = \frac{L(f(\mathbf{x}^1)-f^*)}{dK\gamma^2}$, $\lambda \le \sqrt{\frac{dL\gamma}{72K(f(\mathbf{x}^1)-f^*)}}$, $\|\mathbf{x}^1\|_\infty \le \sqrt{\frac{K(f(\mathbf{x}^1)-f^*)}{dL}}$, $\|\mathbf{x}^k\|_\infty < \frac{1}{\lambda}$, and accordingly*

$$\frac{1}{K}\sum_{k=1}^K \mathbb{E}\left[\|\nabla f(\mathbf{x}^k)\|_1\right] \le 38\sqrt{\frac{dL(f(\mathbf{x}^1)-f^*)}{K\gamma}}.$$

Theorem 1 demonstrates that AdamW minimizes the gradient norm directly while restricting $\|\mathbf{x}^k\|_\infty < \frac{1}{\lambda}$. As a comparison, $\ell_2$ regularized Adam only minimizes $\|\nabla f(\mathbf{x}) + \lambda \mathbf{x}\|$, rather than $\|\nabla f(\mathbf{x})\|$.

As a special case, we also establish the same convergence rate for Adam in the following corollary under slightly relaxed parameter settings. The complete description of Corollary 1 is given in Appendix B.

**Corollary 1** *With the same assumptions and parameter settings of $1 - \theta$, $\eta$, and $\varepsilon$ as Theorem 1, but only requiring $0 \le \beta \le 1$ rather than both $\theta \le \beta \le \sqrt{\theta}$ and $\|\mathbf{x}^1\|_\infty \le \sqrt{\frac{K(f(\mathbf{x}^1)-f^*)}{dL}}$, we have the same convergence rate for Adam as established in Theorem 1.*

## 2.1 Optimality of Our Convergence Rate

When comparing our convergence rate (3) with the optimal rate (4) of SGD, which aligns with the lower bound in nonconvex stochastic optimization, we observe that our rate is also optimal with respect to $K$, $\sigma_s$, $L$, and $f(\mathbf{x}^1) - f^*$. The only remaining uncertainty concerns the tightness of the dimension $d$. Theoretically, $\|\nabla f(\mathbf{x})\|_2 \ll \|\nabla f(\mathbf{x})\|_1 \le \sqrt{d}\|\nabla f(\mathbf{x})\|_2$ holds for any high-dimensional vector $\mathbf{x}$, and when each element of $\nabla f(\mathbf{x})$ is drawn from Gaussian distribution $\mathcal{N}(0,1)$, we have $\mathbb{E}\left[\|\nabla f(\mathbf{x})\|_1\right] \ge \sqrt{\frac{2d}{\pi}}\mathbb{E}\left[\|\nabla f(\mathbf{x})\|_2\right]$ from Lemma 1. Empirically, experiments on real deep neural networks training confirm $\|\nabla f(\mathbf{x})\|_1 = \Theta(\sqrt{d})\|\nabla f(\mathbf{x})\|_2$, as demonstrated in Figure 2. Thus, our convergence rate (3) can be regarded to be analogous to SGD's optimal rate (4).

**Lemma 1** *When each entry of $\mathbf{x} \in \mathbb{R}^d$ is generated from Gaussian distribution with zero mean and unit variance, we have $\mathbb{E}\left[\|\mathbf{x}\|_1\right] \ge \sqrt{\frac{2d}{\pi}}\mathbb{E}\left[\|\mathbf{x}\|_2\right]$.*

---

[1]We gratefully thank the anonymous NeurIPS reviewer to derive this looser bound. Our original bound is $\theta \le \beta \le \frac{(1+\theta)^2}{4}$.

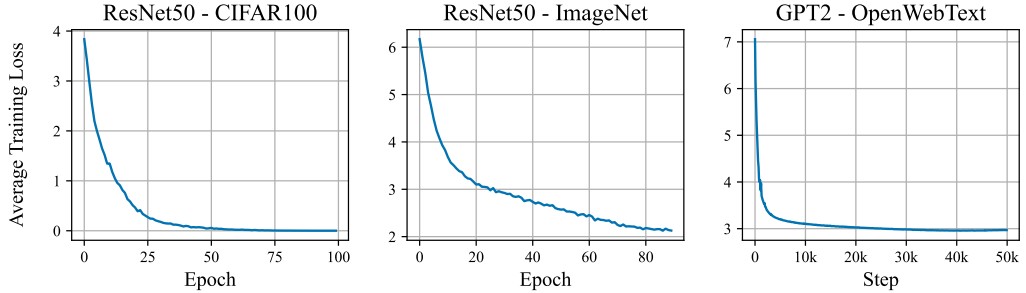

Figure 1: Illustration of average training loss $f(\mathbf{x}^k)$ over epochs/steps, and at the initialization, $f(\mathbf{x}^1) \leq 8$.

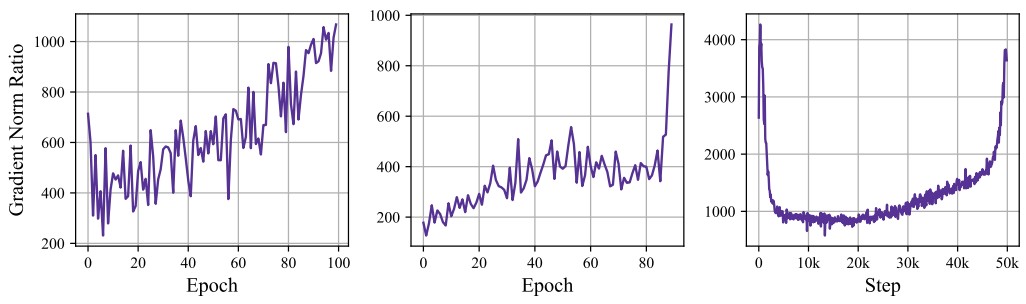

Figure 2: Illustration of $\|\nabla f(\mathbf{x}^k)\|_1 = \Theta(\sqrt{d})\|\nabla f(\mathbf{x}^k)\|_2$ over epochs/steps. The gradient norm ratio shows $\frac{\|\nabla f(\mathbf{x}^k)\|_1}{\|\nabla f(\mathbf{x}^k)\|_2}$, and $\sqrt{d} = 4868$, $5060$, and $11136$, respectively.

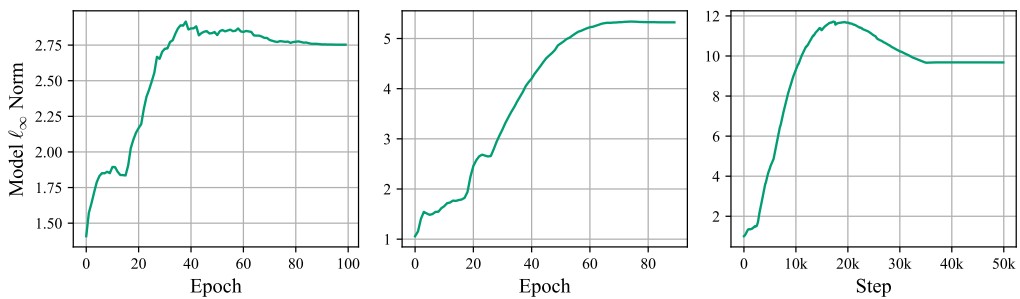

Figure 3: Illustration of $\|\mathbf{x}^k\|_\infty < \frac{1}{\lambda}$ over epochs/steps. The model $\ell_\infty$ norm shows $\|\mathbf{x}^k\|_\infty$, and $\lambda = 0.01$, $0.1$, and $0.05$, respectively.

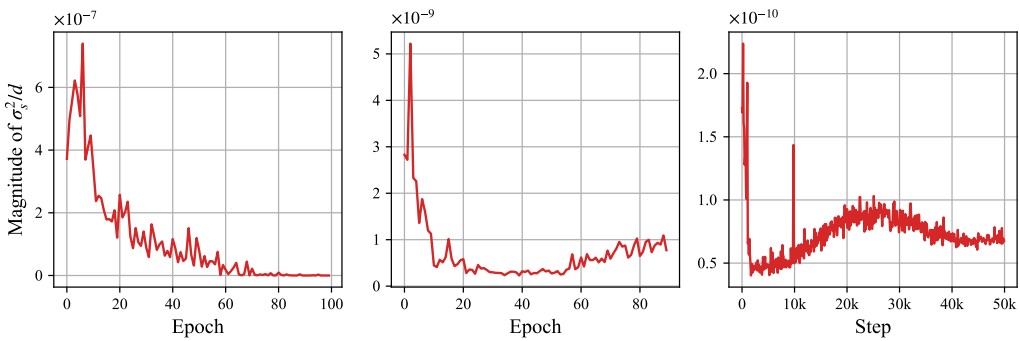

Figure 4: Illustration of small $\frac{\sigma_s^2}{d}$ over epochs/steps. The magnitude $\sigma_s^2$ is approximated by $\|\mathbf{g}^k - \nabla f(\mathbf{x}^k)\|^2$ without taking expectation, and $d = 2.37 \times 10^7$, $2.56 \times 10^7$, and $1.24 \times 10^8$, respectively.

Recently, Jiang et al. [11] established a fundamental lower bound for SGD when measuring gradients by $\ell_1$ norm, which is of order $\Omega\left(\sqrt{\frac{dL(f(\mathbf{x}^1)-f^*)}{K}} + \sqrt[4]{\frac{dL(f(\mathbf{x}^1)-f^*)\|\boldsymbol{\sigma}\|_1^2}{K}}\right)$ under Assumptions 1-3. When $\|\boldsymbol{\sigma}\|_1 \approx \sqrt{d}\|\boldsymbol{\sigma}\|_2 = \sqrt{d}\sigma_s$, this lower bound precisely aligns with our convergence rate in (3). We further conjecture that this lower bound applies more broadly to general first-order stochastic optimization algorithms under $\ell_1$ norm gradient measurement. This would imply that our derived convergence rate is nearly tight.

## 2.2 Separating the Convergence Rate by the Noise Variance

In Theorem 1, we separate the convergence rate by the magnitude of $\sigma_s$. When $\sigma_s^2 \geq \frac{L(f(\mathbf{x}^1)-f^*)}{K\gamma^2}$, both the convergence rates of AdamW and Adam are $\mathcal{O}(\frac{\sqrt{d}}{K^{1/4}})$. When $\sigma_s^2$ becomes smaller than $\frac{L(f(\mathbf{x}^1)-f^*)}{K\gamma^2}$, the convergence rates improve to $\mathcal{O}(\sqrt{\frac{d}{K}})$, matching that of gradient descent measured by $\ell_1$ norm.

## 2.3 Reasonable Weight Decay Parameter and Initialization Interval

In Theorem 1, we set the weight decay parameter $\lambda$ smaller than $\frac{\sqrt{d}}{\sqrt{72}K^{3/4}}\sqrt[4]{\frac{L^3}{\hat{\sigma}_s^2(f(\mathbf{x}^1)-f^*)}}$. In modern deep neural networks, the dimension $d$ is typically extremely large, for example, $d = 1.75 \times 10^{11}$ in GPT-3, making $\frac{\sqrt{d}}{K^{3/4}}$ almost certainly exceed 0.01, which is the default setting of $\lambda$ in PyTorch official implementation. For example, in the experiments of our paper, we train ResNet-50 on i) CIFAR-100 and ii) ImageNet dataset, and GPT-2 on iii) OpenWebText, and observe $(K, d) = (39100, 2.37 \times 10^7)$, $(28080, 2.56 \times 10^7)$, and $(50000, 1.24 \times 10^8)$, resulting in $\frac{\sqrt{d}}{K^{3/4}} \approx 1.75$, $2.33$, and $3.33$, respectively. We empirically show in Appendix D that large $\lambda$ may cause AdamW not converge and thus a upper bound is necessary. We also initialize $\|\mathbf{x}^1\|_\infty \leq \sqrt{\frac{K(f(\mathbf{x}^1)-f^*)}{dL}}$. Although $\sqrt{\frac{K}{d}}$ is typically smaller than 1 in large language models training, it remains not too small. In practical configurations, we often initialize the network weights very small. On the other hand, although we always initialize the scale parameter in BatchNorm/LayerNorm to 1, we do not use weight decay for the scale parameter in practice.

## 2.4 Small $\varepsilon$ Setting

In practice, $\varepsilon$ is typically set to a very small value, for example, approximately $10^{-16}$ in PyTorch implementation[2], to prevent division by zero while maintaining the adaptive properties of AdamW and Adam. Larger $\varepsilon$ values would make AdamW and Adam behave similarly to SGD, losing its adaptive learning rate adjustment. In Theorem 1, we set $\varepsilon = \frac{\hat{\sigma}_s^2}{d} = \max\left\{\frac{\sigma_s^2}{d}, \frac{L(f(\mathbf{x}^1)-f^*)}{dK\gamma^2}\right\}$, which remains small due to extremely large $d$ and modest $\sigma_s^2$. We have empirically shown in Figure 4 that $\frac{\sigma_s^2}{d} \approx 10^{-7}$, $10^{-9}$, and $10^{-10}$ in our experiments of ResNet-50 on CIFAR-100 and ImageNet and GPT-2 on OpenWebText, respectively. Intuitively, $\varepsilon$ should be smaller than the square of stochastic gradient at each coordinate, otherwise, $\varepsilon$ would dominate the magnitude of $\mathbf{v}_i^k$ in $\frac{1}{\sqrt{\mathbf{v}_i^k}+\varepsilon}$. Our setting of $\varepsilon$, the coordinate-wise average of gradient noise variance, approximately resides at this critical threshold. Notably, our convergence rates for both AdamW and Adam do not depend on $\varepsilon$ explicitly. In comparison, existing convergence rates for AdamW and Adam in the literature either explicitly depend on $\varepsilon$ or exhibit a higher dependence on the dimension $d$.

## 2.5 Unpractical Settings of $\eta$, $\theta$, and $\beta$

In Theorem 1, we set the learning rate $\eta$ very small and the parameters $\theta$ and $\beta$ nearly equal to 1 to satisfy the proof requirements. This differs from standard implementations where $(\theta, \beta) = (0.9, 0.999)$ is typically used. Although investigating AdamW/Adam's property under realistic

---

[2]In PyTorch official implementation, $\varepsilon$ appeared in a different place in $\frac{\eta}{\sqrt{\mathbf{v}^k}+\varepsilon}$ and $\varepsilon = 10^{-8}$, while we use $\frac{\eta}{\sqrt{\mathbf{v}^k+\varepsilon}}$.

configurations represents an important research direction, as it could yield valuable insights for deep learning hyperparameters tuning, the practical configurations may not guarantee the convergence. For instance, prior work [14, 15] demonstrates through constructed examples that Adam with common hyperparameters ($\theta = 0.9$, $\beta = \{0.999, 0.997, 0.995, 0.993\}$, and $\eta_k = \frac{0.1}{\sqrt{k}}$) fails to converge to stationary points (see [15, Figuire 2]). On the other hand, empirical evidence from recent studies [16, Figure 9] [17, Figure 6] demonstrate that during the training of language models with practical parameter configurations, the gradient norm hardly decreases during the training run, although the objective function decreases sufficiently.

## 2.6   No Conflict with [6]

For sufficiently large weight decay parameter $\lambda$ where no critical points exist within the constrained domain of problem (1), the KKT conditions (2) serve as a natural convergence metric. As $\lambda$ diminishes, problem (1) asymptotically approaches an unconstrained optimization problem, and AdamW reduces to Adam in the limit. There exists a critical threshold beyond which $\|\nabla f(\mathbf{x})\|_1$ also becomes a viable metric for convergence. Consequently, our results do not conflict with [6].

## 3   Proof Sketch

In this section, we outline the proof sketch of Theorem 1. The detailed proofs are provided in Appendix A. From the Lipschitz smoothness of $f(\mathbf{x})$ and the update of $\mathbf{x}^{k+1}$ in Algorithm 1, we have

$$
\mathbb{E}_k\left[f(\mathbf{x}^{k+1})|\mathcal{F}_{k-1}\right] - f(\mathbf{x}^k)
$$

$$
\leq \mathbb{E}_k\left[\langle\nabla f(\mathbf{x}^k), \mathbf{x}^{k+1} - \mathbf{x}^k\rangle + \frac{L}{2}\|\mathbf{x}^{k+1} - \mathbf{x}^k\|^2\Big|\mathcal{F}_{k-1}\right]
$$

$$
= \mathbb{E}_k\left[\underbrace{-\eta\sum_{i=1}^{d}\left\langle\nabla_i f(\mathbf{x}^k), \frac{\mathbf{m}_i^k + \lambda\mathbf{x}_i^k\sqrt{\mathbf{v}_i^k + \varepsilon}}{\sqrt{\mathbf{v}_i^k + \varepsilon}}\right\rangle}_{\text{term (a)}} + \underbrace{\frac{L\eta^2}{2}\sum_{i=1}^{d}\frac{\left|\mathbf{m}_i^k + \lambda\mathbf{x}_i^k\sqrt{\mathbf{v}_i^k + \varepsilon}\right|^2}{\mathbf{v}_i^k + \varepsilon}}_{\text{term (b)}}\Big|\mathcal{F}_{k-1}\right]. \tag{5}
$$

Decompose term (a) into

$$
-\frac{\eta}{2}\sum_{i=1}^{d}\frac{\left|\nabla_i f(\mathbf{x}^k)\right|^2}{\sqrt{\mathbf{v}_i^k + \varepsilon}} \underbrace{-\frac{\eta}{2}\sum_{i=1}^{d}\frac{\left|\mathbf{m}_i^k + \lambda\mathbf{x}_i^k\sqrt{\mathbf{v}_i^k + \varepsilon}\right|^2}{\sqrt{\mathbf{v}_i^k + \varepsilon}}}_{\text{term (c)}} + \underbrace{\frac{\eta}{2}\sum_{i=1}^{d}\frac{\left|\nabla_i f(\mathbf{x}^k) - \mathbf{m}_i^k - \lambda\mathbf{x}_i^k\sqrt{\mathbf{v}_i^k + \varepsilon}\right|^2}{\sqrt{\mathbf{v}_i^k + \varepsilon}}}_{\text{term (d)}} \tag{6}
$$

and relax term (b) as follows to absorb it within term (c)

$$
\text{term (b)} \leq \frac{L\eta^2}{2\sqrt{\varepsilon}}\sum_{i=1}^{d}\frac{\left|\mathbf{m}_i^k + \lambda\mathbf{x}_i^k\sqrt{\mathbf{v}_i^k + \varepsilon}\right|^2}{\sqrt{\mathbf{v}_i^k + \varepsilon}} \overset{\eta\leq\frac{\sqrt{\varepsilon}}{2L}}{\leq} \frac{\eta}{4}\sum_{i=1}^{d}\frac{\left|\mathbf{m}_i^k + \lambda\mathbf{x}_i^k\sqrt{\mathbf{v}_i^k + \varepsilon}\right|^2}{\sqrt{\mathbf{v}_i^k + \varepsilon}}. \tag{7}
$$

Next, we consider term (d) and relax it as follows

$$
\text{term (d)} \leq \frac{\eta}{\sqrt{\varepsilon}}\left\|\nabla f(\mathbf{x}^k) - \mathbf{m}^k\right\|^2 + \eta\sum_{i=1}^{d}|\lambda\mathbf{x}_i^k|^2\sqrt{\mathbf{v}_i^k + \varepsilon}. \tag{8}
$$

We see that the parameter $\varepsilon$ plays a pivotal rule in steps (7) and (8). Combing (5)-(8), we have

$$
\mathbb{E}_k\left[f(\mathbf{x}^{k+1})|\mathcal{F}_{k-1}\right] - f(\mathbf{x}^k) \leq \mathbb{E}_k\left[-\frac{\eta}{2}\sum_{i=1}^{d}\frac{\left|\nabla_i f(\mathbf{x}^k)\right|^2}{\sqrt{\mathbf{v}_i^k + \varepsilon}} - \frac{\eta}{4}\sum_{i=1}^{d}\frac{\left|\mathbf{m}_i^k + \lambda\mathbf{x}_i^k\sqrt{\mathbf{v}_i^k + \varepsilon}\right|^2}{\sqrt{\mathbf{v}_i^k + \varepsilon}}\right.
$$

$$
\left. + \underbrace{\frac{\eta}{\sqrt{\varepsilon}}\left\|\nabla f(\mathbf{x}^k) - \mathbf{m}^k\right\|^2}_{\text{term (e)}} + \eta\sum_{i=1}^{d}|\lambda\mathbf{x}_i^k|^2\sqrt{\mathbf{v}_i^k + \varepsilon}\Big|\mathcal{F}_{k-1}\right]. \tag{9}
$$

Considering term (e), we can use standard techniques in the analysis of momentum SGD to build a recursion (Lemma 4) as follows

$$\mathbb{E}_k \left[ \left\| \nabla f(\mathbf{x}^k) - \mathbf{m}^k \right\|^2 |\mathcal{F}_{k-1} \right]$$

$$\leq \theta \left\| \mathbf{m}^{k-1} - \nabla f(\mathbf{x}^{k-1}) \right\|^2 + \frac{L^2 \eta^2}{\sqrt{\varepsilon}(1-\theta)} \sum_{i=1}^{d} \frac{\left| \mathbf{m}_i^{k-1} + \lambda \mathbf{x}_i^{k-1} \sqrt{\mathbf{v}_i^{k-1} + \varepsilon} \right|^2}{\sqrt{\mathbf{v}_i^{k-1} + \varepsilon}} + (1-\theta)^2 \sigma_s^2. \tag{10}$$

Multiplying both sides of (10) by $\frac{\eta}{\sqrt{\varepsilon}(1-\theta)}$, adding it to (9), and letting $\eta^2 \leq \frac{\varepsilon(1-\theta)^2}{4L^2}$, we have

$$\mathbb{E}_k \left[ f(\mathbf{x}^{k+1}) - f^* + \frac{\eta\theta}{\sqrt{\varepsilon}(1-\theta)} \left\| \nabla f(\mathbf{x}^k) - \mathbf{m}^k \right\|^2 + \frac{\eta}{4} \sum_{i=1}^{d} \frac{\left| \mathbf{m}_i^k + \lambda \mathbf{x}_i^k \sqrt{\mathbf{v}_i^k + \varepsilon} \right|^2}{\sqrt{\mathbf{v}_i^k + \varepsilon}} \Big| \mathcal{F}_{k-1} \right]$$

$$\leq f(\mathbf{x}^k) - f^* + \sum_{i=1}^{d} \mathbb{E}_k \left[ \underbrace{-\frac{\eta}{2} \frac{\left| \nabla_i f(\mathbf{x}^k) \right|^2}{\sqrt{\mathbf{v}_i^k + \varepsilon}}}_{\text{term (f)}} + \underbrace{\eta |\lambda \mathbf{x}_i^k|^2 \sqrt{\mathbf{v}_i^k + \varepsilon}}_{\text{term (g)}} \Big| \mathcal{F}_{k-1} \right] \tag{11}$$

$$+ \frac{\eta\theta}{\sqrt{\varepsilon}(1-\theta)} \left\| \nabla f(\mathbf{x}^{k-1}) - \mathbf{m}^{k-1} \right\|^2 + \frac{\eta}{4} \sum_{i=1}^{d} \frac{\left| \mathbf{m}_i^{k-1} + \lambda \mathbf{x}_i^{k-1} \sqrt{\mathbf{v}_i^{k-1} + \varepsilon} \right|^2}{\sqrt{\mathbf{v}_i^{k-1} + \varepsilon}} + \frac{\eta(1-\theta)\sigma_s^2}{\sqrt{\varepsilon}}.$$

The above analysis comes from the standard framework and contains nothing new. We can recursively eliminate certain terms in (11) after telescoping, except for the troublesome term (g). The following outlines the key technical components of our proof to address term (g) and achieve the tight convergence rate.

### 3.1  Bounding $|\lambda \mathbf{x}_i^k|$ in Term (g) by $O(\frac{1}{K^{1/4}})$

The analysis for AdamW proves more challenging than for SignSGD-type methods with weight decay [18], because SignSGD maintains a fixed update size 1, whereas AdamW's updates can be arbitrarily large. Specifically, for AdamW and Adam, we have $\frac{|\mathbf{m}_i^k|^2}{\mathbf{v}_i^k} \leq \frac{(1-\theta)^2 \beta}{(1-\beta)(\beta-\theta^2)}$ (Lemma 2), where the latter is minimized to be 1 by setting $\theta = \beta$. However, when setting $\theta = O(1)$ (for example, $\theta = 0.9$) and $\beta = 1 - \frac{1}{K}$, we have $\frac{(1-\theta)^2 \beta}{(1-\beta)(\beta-\theta^2)} = O(K)$, leading to unbounded updates in AdamW. This fundamental difficulty prevents direct extension of the proof framework in [10] to AdamW. We set $\theta \leq \beta \leq \sqrt{\theta}$ in Theorem 1 such that $\frac{(1-\theta)^2 \beta}{(1-\beta)(\beta-\theta^2)} \leq 4$ (Lemma 2). Then for the update of $\mathbf{x}^{k+1}$ in AdamW, we have

$$\|\mathbf{x}^{k+1}\|_\infty - \frac{2}{\lambda} \leq (1-\eta\lambda)^k \left( \|\mathbf{x}^1\|_\infty - \frac{2}{\lambda} \right).$$

When $(1-\eta\lambda)^k$ decreases fast, we have $(1-\eta\lambda)^k \left( \|\mathbf{x}^1\|_\infty - \frac{2}{\lambda} \right) \to 0$ and $\|\mathbf{x}^{k+1}\|_\infty$ is loosely bounded by $\frac{2}{\lambda}$, which is far from our target $\lambda \|\mathbf{x}^{k+1}\|_\infty \leq O(\frac{1}{K^{1/4}})$. To address this issue, we control the decrease of $(1-\eta\lambda)^k$ by setting parameter $\lambda$ properly such that $\eta\lambda \leq \frac{\sqrt{\nu}}{2K^{5/4}}$ and $(1-\eta\lambda)^k \geq e^{-\frac{\sqrt{\nu}}{K^{1/4}}} \geq 1 - \frac{\sqrt{\nu}}{K^{1/4}}$ for some $\nu$ and any $k \leq K$. Equipped with proper initialization of $\|\mathbf{x}^1\|_\infty \leq \frac{\sqrt{\nu}}{K^{1/4}\lambda}$, we finally have (Lemma 3)

$$\|\mathbf{x}^{k+1}\|_\infty \leq \frac{2}{\lambda} - \left(1 - \frac{\sqrt{\nu}}{K^{1/4}}\right) \left(\frac{2}{\lambda} - \frac{\sqrt{\nu}}{K^{1/4}\lambda}\right) \leq \frac{3}{\lambda} \frac{\sqrt{\nu}}{K^{1/4}},$$

and

$$\text{term (g)} \leq \frac{9\eta\nu}{K^{1/2}} \sqrt{\mathbf{v}_i^k + \varepsilon}.$$

Intuitively, when the initialization is far from the boundary of problem (1) and $(1-\eta\lambda)^k \approx 1$, the iterates $\mathbf{x}^{k+1}$ are guaranteed to be far from the boundary throughout the optimization process.

## 3.2 Absorbing Term (g) within Term (f)

To absorb term (g) within term (f), we first relax $\sqrt{\mathbf{v}_i^k + \varepsilon}$ to $\sqrt{\widetilde{\mathbf{v}}_i^k + \varepsilon}$ as follows by the concavity of $\sqrt{x}$ and $-\frac{1}{\sqrt{x}}$

$$\mathbb{E}_k\left[\underbrace{-\frac{\eta}{2}\frac{|\nabla_i f(\mathbf{x}^k)|^2}{\sqrt{\mathbf{v}_i^k + \varepsilon}}}_{\text{term (f)}} + \underbrace{\eta|\lambda \mathbf{x}_i^k|^2 \sqrt{\mathbf{v}_i^k + \varepsilon}}_{\text{term (g)}}\Big| \mathcal{F}_{k-1}\right] \le -\frac{\eta}{2}\frac{|\nabla_i f(\mathbf{x}^k)|^2}{\sqrt{\widetilde{\mathbf{v}}_i^k + \varepsilon}} + \frac{9\eta\nu}{K^{1/2}}\sqrt{\widetilde{\mathbf{v}}_i^k + \varepsilon},$$

where we define $\widetilde{\mathbf{v}}_i^k = \beta \mathbf{v}_i^{k-1} + (1 - \beta)\left(|\nabla_i f(\mathbf{x}^k)|^2 + \sigma_i^2\right)$. Then, we can bound $\sqrt{\widetilde{\mathbf{v}}_i^k + \varepsilon}$ as follows (Lemma 5) and absorb term (h) within $-\frac{\eta}{2}\sum_{k=1}^K \sum_{i=1}^d \frac{|\nabla_i f(\mathbf{x}^k)|^2}{\sqrt{\widetilde{\mathbf{v}}_i^k + \varepsilon}}$ derived from term (f),

$$\sum_{k=1}^K \sum_{i=1}^d \mathbb{E}_{\mathcal{F}_{k-1}}\left[\sqrt{\widetilde{\mathbf{v}}_i^k + \varepsilon}\right] \le K\|\boldsymbol{\sigma}\|_1 + Kd\sqrt{\varepsilon} + 2\underbrace{\sum_{k=1}^K \sum_{i=1}^d \mathbb{E}_{\mathcal{F}_{t-1}}\left[\frac{|\nabla_i f(\mathbf{x}^k)|^2}{\sqrt{\widetilde{\mathbf{v}}_i^k + \varepsilon}}\right]}_{\text{term (h)}}. \tag{12}$$

Summing (11) over $k$ and combing the above analysis, we have

$$\mathbb{E}_{\mathcal{F}_K}\left[f(\mathbf{x}^{K+1}) - f^* + \frac{\eta\theta}{\sqrt{\varepsilon}(1 - \theta)}\|\nabla f(\mathbf{x}^K) - \mathbf{m}^K\|^2 + \frac{\eta}{4}\sum_{i=1}^d \frac{|\mathbf{m}_i^K + \lambda \mathbf{x}_i^K \sqrt{\mathbf{v}_i^K + \varepsilon}|^2}{\sqrt{\mathbf{v}_i^K + \varepsilon}}\right]$$

$$\le -\frac{\eta}{2}\sum_{k=1}^K \sum_{i=1}^d \mathbb{E}_{\mathcal{F}_{k-1}}\left[\frac{|\nabla_i f(\mathbf{x}^k)|^2}{\sqrt{\widetilde{\mathbf{v}}_i^k + \varepsilon}}\right] + \frac{9\eta\nu}{K^{1/2}}\left(\underbrace{K\|\boldsymbol{\sigma}\|_1 + Kd\sqrt{\varepsilon}}_{\text{term (i)}} + 2\sum_{k=1}^K \sum_{i=1}^d \mathbb{E}_{\mathcal{F}_{k-1}}\left[\frac{|\nabla_i f(\mathbf{x}^k)|^2}{\sqrt{\widetilde{\mathbf{v}}_i^k + \varepsilon}}\right]\right)$$

$$+ \underbrace{f(\mathbf{x}^1) - f^* + \frac{\eta}{\sqrt{\varepsilon}(1 - \theta)}\mathbb{E}_{\mathcal{F}_1}\left[\|\nabla f(\mathbf{x}^1) - \mathbf{m}^1\|^2\right] + \frac{K\eta(1 - \theta)\sigma_s^2}{\sqrt{\varepsilon}}}_{\text{term (j)}}$$

$$\le -\frac{\eta}{4}\sum_{k=1}^K \sum_{i=1}^d \mathbb{E}_{\mathcal{F}_{k-1}}\left[\frac{|\nabla_i f(\mathbf{x}^k)|^2}{\sqrt{\widetilde{\mathbf{v}}_i^k + \varepsilon}}\right] + 18\eta\nu d\sqrt{K\varepsilon} + \text{term (j)}$$

by letting $\frac{9\nu}{K^{1/2}} \le \frac{1}{8}$ and $\varepsilon = \frac{\sigma_s^2}{d}$ such that $K\|\boldsymbol{\sigma}\|_1 \le Kd\sqrt{\varepsilon}$. Letting $\nu = \frac{1}{72d}\sqrt{\frac{\sigma_s^2 L(f(\mathbf{x}^1) - f^*)}{\varepsilon^2}}$, $1 - \theta = \sqrt{\frac{L(f(\mathbf{x}^1) - f^*)}{K\sigma_s^2}}$ and $\eta = \sqrt{\frac{\varepsilon(f(\mathbf{x}^1) - f^*)}{4K\sigma_s^2 L}}$, both term (j) and $\eta\nu d\sqrt{K\varepsilon}$ are of the order $\eta\sqrt{\frac{K\sigma_s^2 L(f(\mathbf{x}^1) - f^*)}{\varepsilon}}$. This accounts for why bounding $|\lambda \mathbf{x}_i^k|$ by $\mathcal{O}(\frac{\sqrt{\nu}}{K^{1/4}})$, otherwise, term (i) would slow the convergence rate established in Theorem 1. Intuitively, when $\nabla_i f(\mathbf{x}^k) \approx 0$ such that $\widetilde{\mathbf{v}}_i^k = \beta^k \mathbf{v}_i^0 + (1 - \beta)\sum_{r=1}^k \beta^{k-r}\left(|\nabla_i f(\mathbf{x}^r)|^2 + \sigma_i^2\right) \approx \sigma_i^2$, we have $\sum_{k=1}^K \sum_{i=1}^d \sqrt{\widetilde{\mathbf{v}}_i^k + \varepsilon} \approx K\|\boldsymbol{\sigma}\|_1 + Kd\sqrt{\varepsilon}$, making term (i) non-negligible in (12).

## 3.3 Eliminating $\varepsilon$ in the Final Convergence Rate

Based on the above analysis, we get the following bound

$$\sum_{k=1}^K \sum_{i=1}^d \mathbb{E}_{\mathcal{F}_{k-1}}\left[\frac{|\nabla_i f(\mathbf{x}^k)|^2}{\sqrt{\widetilde{\mathbf{v}}_i^k + \varepsilon}}\right] \le \mathcal{O}\left(\sqrt{\frac{K\sigma_s^2 L(f(\mathbf{x}^1) - f^*)}{\varepsilon}}\right).$$

Using Holder's inequality and (12) again, we finally have

$$\left(\sum_{k=1}^K \mathbb{E}_{\mathcal{F}_{k-1}}\left[\|\nabla f(\mathbf{x}^k)\|_1\right]\right)^2$$

$$\le \left(\sum_{k=1}^K \sum_{i=1}^d \mathbb{E}_{\mathcal{F}_{k-1}}\left[\frac{|\nabla_i f(\mathbf{x}^k)|^2}{\sqrt{\widetilde{\mathbf{v}}_i^k + \varepsilon}}\right]\right)\left(\sum_{k=1}^K \sum_{i=1}^d \mathbb{E}_{\mathcal{F}_{k-1}}\left[\sqrt{\widetilde{\mathbf{v}}_i^k + \varepsilon}\right]\right)$$

$$\leq \left( \sum_{k=1}^{K} \sum_{i=1}^{d} \mathbb{E}_{\mathcal{F}_{k-1}} \left[ \frac{|\nabla_i f(\mathbf{x}^k)|^2}{\sqrt{\widetilde{\mathbf{v}}_i^k} + \varepsilon} \right] \right) \left( K\|\boldsymbol{\sigma}\|_1 + Kd\sqrt{\varepsilon} + 2 \sum_{k=1}^{K} \sum_{i=1}^{d} \mathbb{E}_{\mathcal{F}_{k-1}} \left[ \frac{|\nabla_i f(\mathbf{x}^k)|^2}{\sqrt{\widetilde{\mathbf{v}}_i^k} + \varepsilon} \right] \right)$$

$$\leq \mathcal{O} \left( \sqrt{\frac{K\sigma_s^2 L(f(\mathbf{x}^1) - f^*)}{\varepsilon}} \left( \sqrt{\frac{K\sigma_s^2 L(f(\mathbf{x}^1) - f^*)}{\varepsilon}} + K\|\boldsymbol{\sigma}\|_1 + Kd\sqrt{\varepsilon} \right) \right)$$

and

$$\frac{1}{K} \sum_{k=1}^{K} \mathbb{E}_{\mathcal{F}_{k-1}} \left[ \|\nabla f(\mathbf{x}^k)\|_1 \right]$$

$$\leq \mathcal{O} \left( \frac{1}{K} \left( \sqrt{\frac{K\sigma_s^2 L(f(\mathbf{x}^1) - f^*)}{\varepsilon}} + \underbrace{\sqrt[4]{\frac{K\sigma_s^2 L(f(\mathbf{x}^1) - f^*)}{\varepsilon} \left( K\|\boldsymbol{\sigma}\|_1 + Kd\sqrt{\varepsilon} \right)^2}}_{\text{term (k)}} \right) \right).$$

The above convergence rate is not optimal due to its explicit dependence on $\varepsilon$, which is absent from the optimal rate (4) of SGD. By setting $\varepsilon = \frac{\sigma_s^2}{d}$, we obtain $K\|\boldsymbol{\sigma}\|_1 \leq Kd\sqrt{\varepsilon}$ and $(K\|\boldsymbol{\sigma}\|_1 + Kd\sqrt{\varepsilon})^2 \leq 4K^2 d^2 \varepsilon$, which allows us to eliminate $\varepsilon$ in the denominator of term (k). This yields the following final convergence rate

$$\frac{1}{K} \sum_{k=1}^{K} \mathbb{E}_{\mathcal{F}_{k-1}} \left[ \|\nabla f(\mathbf{x}^k)\|_1 \right] \leq \mathcal{O} \left( \sqrt{\frac{dL(f(\mathbf{x}^1) - f^*)}{K}} + \frac{\sqrt{d}}{K^{1/4}} \sqrt[4]{\sigma_s^2 L(f(\mathbf{x}^1) - f^*)} \right).$$

Although smaller value of $\varepsilon$ does not affect the convergence of AdamW and Adam, this term cannot be eliminated any more and consequently slows the convergence rate by introducing explicit $\varepsilon$-dependence. On the other hand, while larger $\varepsilon$ does not impact the convergence rate, it makes AdamW closer to SGD.

At last, in order to incorporate the scenario when $\sigma_s^2 \leq \frac{L(f(\mathbf{x}^1) - f^*)}{K}$, we define $\hat{\sigma}_s^2 = \max\left\{ \sigma_s^2, \frac{L(f(\mathbf{x}^1) - f^*)}{K\gamma^2} \right\}$ with any constant $\gamma \in (0, 1]$ and replace $\sigma_s^2$ by $\hat{\sigma}_s^2$ in the definitions of $\varepsilon, \nu, 1 - \theta$, and $\eta$.

## 4 Literature Comparisons

In this section, we compare our theoretical results with representative ones in the literature. A substantial amount of literature exists regarding the convergence analysis of adaptive gradient algorithms, such as [19, 20, 21, 22, 23] for AdaGrad-norm, [22, 11, 12, 24, 25] for AdaGrad, [26, 27, 28, 10, 13] for RMSProp, [29] for Adam-norm, [30, 26, 27, 31, 32, 14, 33, 15, 34, 35, 36, 37, 38, 39] for Adam, and [40, 41, 42, 43, 44, 45, 46, 47, 48, 49] for other variants. We primarily compare with the literature on AdamW and Adam. For Adam, we restrict our comparison to studies with the state-of-the-art convergence rates that do not require the bounded gradient assumption.

### 4.1 AdamW: Comparison with [7]

To the best of our knowledge based on a comprehensive literature review, [7] appears to be the only existing paper addressing AdamW's convergence and convergence rate. We compare with [7] in the following aspects. Firstly, the assumptions in [7] are stronger than ours. Denoting $f(\mathbf{x}) = \mathbb{E}_{\zeta \in D}[f(\mathbf{x}; \zeta)]$, they assumed $\|\nabla f(\mathbf{y}; \zeta) - \nabla f(\mathbf{x}; \zeta)\| \leq L\|\mathbf{y} - \mathbf{x}\|$ (under which the lower bound is $\mathcal{O}(\frac{1}{\epsilon^3})$, rather than $\mathcal{O}(\frac{1}{\epsilon^4})$ [9]) and $\|\mathbf{g}^k\|_\infty \leq c_\infty$, while we only assume $\|\nabla f(\mathbf{y}) - \nabla f(\mathbf{x})\| \leq L\|\mathbf{y} - \mathbf{x}\|$ without the bounded gradient assumption. Secondly, they set the weight decay parameter $\lambda_k = \lambda(1 - \frac{\beta c_\infty^2}{\varepsilon})^k$, which decreases exponentially, making AdamW reduce to standard Adam in the limit. Thirdly, they establish the complexity of $\mathcal{O}(\max\{\frac{c_\infty^{2.5} L\sigma_s^2 (f(\mathbf{x}^1) - f^*)}{\varepsilon^{1.25}\epsilon^4}, \frac{c_\infty^2 \sigma_s^4}{\varepsilon\epsilon^4}\})$ to achieve $\frac{1}{K} \sum_{k=1}^{K} \mathbb{E}[\|\nabla F_k(\mathbf{x}^k)\|^2] \leq \epsilon^2$, where $F_k$ is a dynamic $\ell_2$ regularized objective. Their complexity depends on $\varepsilon$ explicitly, which is usually small in practice, for example, $\varepsilon \approx 10^{-16}$ in PyTorch implementation. As a comparison, our convergence rate does not depend on $\varepsilon$ explicitly.

## 4.2 Adam: Comparison with [10]

Li et al. [10] studied RMSProp and its momentum extension, where RMSProp is a special case of Adam by letting $\theta = 0$ and $\lambda = 0$ in Algorithm 1. The convergence analysis of Adam presents substantially greater challenges than RMSProp and we cannot extend the proofs in [10] to Adam. Alternatively, this paper uses a different proof framework to establish for Adam the same convergence rate achieved by [10] under identical assumptions. As a trade-off, one limitation of our proof is that it relies on a larger value of parameter $\varepsilon$, although $\varepsilon = \frac{\hat{\sigma}_s^2}{d}$ is very small in practice. Specifically, under the parameter settings of $\beta = 1 - \frac{1}{K}$, $\mathbf{v}_i^0 = \lambda \max\{\sigma_i^2, \frac{1}{dK}\}$, and $\lambda \geq \frac{\sigma_s^2}{KL(f(\mathbf{x}^1)-f^*)}$ in [10], we have $\frac{1}{e^2} \leq \beta^t \leq 1$ for any $t \leq K$ and

$$\mathbf{v}_i^k = \beta^k \mathbf{v}_i^0 + (1-\beta)\sum_{t=1}^k \beta^{k-t}|\mathbf{g}_i^t|^2 \approx \frac{\sigma_i^2}{K}\frac{\sigma_s^2}{L(f(\mathbf{x}^1)-f^*)} + \frac{1}{K}\sum_{t=1}^k |\mathbf{g}_i^t|^2,$$

where $\beta^k \mathbf{v}_i^0$ plays the role of $\varepsilon$ in Algorithm 1, which is of the order $\frac{\sigma_i^2}{K}$, or approximately $\frac{\sigma_s^2}{dK}$. As a comparison, in this paper, we have

$$\mathbf{v}_i^k + \varepsilon = \varepsilon + (1-\beta)\sum_{t=1}^k \beta^{k-t}|\mathbf{g}_i^t|^2 = \frac{\hat{\sigma}_s^2}{d} + (1-\beta)\sum_{t=1}^k \beta^{k-t}|\mathbf{g}_i^t|^2 \approx \sigma_i^2 + (1-\beta)\sum_{t=1}^k \beta^{k-t}|\mathbf{g}_i^t|^2.$$

When $\nabla f(\mathbf{x}^t) \approx 0$ such that $|\mathbf{g}_i^t| \approx \sigma_i$, we have $(1-\beta)\sum_{t=1}^k \beta^{k-t}|\mathbf{g}_i^t|^2 \approx \sigma_i^2$. Thus, $\varepsilon$ accounts for nearly half of $(\mathbf{v}_i^k + \varepsilon)$'s size, while in [10], $\beta^k \mathbf{v}_i^0$ only makes up close to $\frac{1}{k}$ of $\mathbf{v}_i^k$'s total size. Other representative studies [34, 35, 33] have derived convergence guarantees for Adam built upon weak $\varepsilon$-dependent analysis. However, these results all yield slower convergence rates than ours with a higher dependence on the dimension $d$.

## 4.3 Adam: Comparison with [37]

Li et al. [37] studied Adam under assumption $\|\mathbf{g}^k - \nabla f(\mathbf{x}^k)\| \leq \sigma_s$ with probability 1 and proved $\frac{1}{K}\sum_{k=1}^K \|\nabla f(\mathbf{x}^k)\|_2^2 \leq \epsilon^2$ with high probability within $\mathcal{O}(\frac{G^{2.5}\sigma_s^2 L(f(\mathbf{x}^1)-f^*)}{\tilde{\varepsilon}^{2.5}\epsilon^4})$ iterations. That is, $\frac{1}{K}\sum_{k=1}^K \|\nabla f(\mathbf{x}^k)\|_2 \leq (\frac{G}{\tilde{\varepsilon}})^{5/8}\frac{1}{K^{1/4}}\sqrt[4]{\sigma_s^2 L(f(\mathbf{x}^1)-f^*)}$, where $G \geq \max\{\tilde{\varepsilon}, \sigma_s, \sqrt{L(f(\mathbf{x}^1)-f^*)}\}$ and $\tilde{\varepsilon}$ appeared in a different place in $\frac{\mathbf{m}^k}{\sqrt{\mathbf{v}^k}+\tilde{\varepsilon}}$ (hence we may consider $\tilde{\varepsilon}$ to be equal to $\sqrt{\varepsilon}$). When $\|\nabla f(\mathbf{x})\|_1 = \Theta(\sqrt{d})\|\nabla f(\mathbf{x})\|_2$, as empirically observed in real-world deep learning training, our convergence rate is $(\frac{G}{\tilde{\varepsilon}})^{5/8}$ times faster than [37]. In PyTorch implementation, the default value of $\tilde{\varepsilon}$ is typically set to $10^{-8}$. To eliminate the dependence on $\varepsilon$, Li et al. [37] requires $\tilde{\varepsilon}^2(\approx \varepsilon) = G^2 \geq \max\{\sigma_s^2, L(f(\mathbf{x}^1)-f^*)\} \geq \sigma_s^2$, while we only need $\varepsilon = \frac{\sigma_s^2}{d}$, which is $d$ times smaller.

## Conclusion

This paper studies the popular AdamW optimizer in deep learning. We establish the convergence rate $\frac{1}{K}\sum_{k=1}^K \mathbb{E}\left[\|\nabla f(\mathbf{x}^k)\|_1\right] \leq \mathcal{O}(\frac{\sqrt{d}C}{K^{1/4}})$ for AdamW measured by $\ell_1$ norm. It can be considered to be analogous to the optimal rate of SGD in the ideal case of $\|\nabla f(\mathbf{x})\|_1 = \Theta(\sqrt{d})\|\nabla f(\mathbf{x})\|_2$, which is verified on real-world deep learning tasks.

An important direction for future research would be to investigate the optimal convergence rate using weak $\varepsilon$-dependent analysis (for example, $\log\frac{1}{\varepsilon}$) for AdamW and Adam. On the other hand, it is currently unclear whether our upper bound on $\lambda$ is tight. Investigating how to prove the optimal convergence rate under a looser upper bound would be meaningful. This study is primarily concerned with theoretical analysis and it does not yield direct negative societal impacts.

## Acknowledgements

H. Li was supported by the NSF China (No. 62476142) and Z. Lin was supported by the NSF China (No. 62276004). Li and Lin are the corresponding authors.

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

# A Proof of Theorem 1

**Proof 1** *As the gradient is L-Lipschitz, we have*

$$
\mathbb{E}_k\left[f(\mathbf{x}^{k+1})|\mathcal{F}_{k-1}\right] - f(\mathbf{x}^k)
$$

$$
\leq \mathbb{E}_k\left[\left\langle \nabla f(\mathbf{x}^k), \mathbf{x}^{k+1} - \mathbf{x}^k\right\rangle + \frac{L}{2}\|\mathbf{x}^{k+1} - \mathbf{x}^k\|^2 \Big| \mathcal{F}_{k-1}\right]
$$

$$
= \mathbb{E}_k\left[-\eta\sum_{i=1}^d \left\langle \nabla_i f(\mathbf{x}^k), \frac{\mathbf{m}_i^k + \lambda\mathbf{x}_i^k\sqrt{\mathbf{v}_i^k + \varepsilon}}{\sqrt{\mathbf{v}_i^k + \varepsilon}}\right\rangle + \frac{L\eta^2}{2}\sum_{i=1}^d \frac{\left|\mathbf{m}_i^k + \lambda\mathbf{x}_i^k\sqrt{\mathbf{v}_i^k + \varepsilon}\right|^2}{\mathbf{v}_i^k + \varepsilon}\Big| \mathcal{F}_{k-1}\right]
$$

$$
= \mathbb{E}_k\left[-\frac{\eta}{2}\sum_{i=1}^d \frac{\left|\nabla_i f(\mathbf{x}^k)\right|^2}{\sqrt{\mathbf{v}_i^k + \varepsilon}} - \frac{\eta}{2}\sum_{i=1}^d \frac{\left|\mathbf{m}_i^k + \lambda\mathbf{x}_i^k\sqrt{\mathbf{v}_i^k + \varepsilon}\right|^2}{\sqrt{\mathbf{v}_i^k + \varepsilon}}\right.
$$

$$
\left. + \frac{\eta}{2}\sum_{i=1}^d \frac{\left|\nabla_i f(\mathbf{x}^k) - \mathbf{m}_i^k - \lambda\mathbf{x}_i^k\sqrt{\mathbf{v}_i^k + \varepsilon}\right|^2}{\sqrt{\mathbf{v}_i^k + \varepsilon}} + \frac{L\eta^2}{2}\sum_{i=1}^d \frac{\left|\mathbf{m}_i^k + \lambda\mathbf{x}_i^k\sqrt{\mathbf{v}_i^k + \varepsilon}\right|^2}{\mathbf{v}_i^k + \varepsilon}\Big| \mathcal{F}_{k-1}\right]
$$

$$
\leq \mathbb{E}_k\left[-\frac{\eta}{2}\sum_{i=1}^d \frac{\left|\nabla_i f(\mathbf{x}^k)\right|^2}{\sqrt{\mathbf{v}_i^k + \varepsilon}} - \frac{\eta}{2}\sum_{i=1}^d \frac{\left|\mathbf{m}_i^k + \lambda\mathbf{x}_i^k\sqrt{\mathbf{v}_i^k + \varepsilon}\right|^2}{\sqrt{\mathbf{v}_i^k + \varepsilon}}\right. \tag{13}
$$

$$
\left. + \eta\sum_{i=1}^d \frac{\left|\nabla_i f(\mathbf{x}^k) - \mathbf{m}_i^k\right|^2 + \left|\lambda\mathbf{x}_i^k\sqrt{\mathbf{v}_i^k + \varepsilon}\right|^2}{\sqrt{\mathbf{v}_i^k + \varepsilon}} + \frac{L\eta^2}{2\sqrt{\varepsilon}}\sum_{i=1}^d \frac{\left|\mathbf{m}_i^k + \lambda\mathbf{x}_i^k\sqrt{\mathbf{v}_i^k + \varepsilon}\right|^2}{\sqrt{\mathbf{v}_i^k + \varepsilon}}\Big| \mathcal{F}_{k-1}\right]
$$

$$
\overset{(1)}{\leq} \mathbb{E}_k\left[-\frac{\eta}{2}\sum_{i=1}^d \frac{\left|\nabla_i f(\mathbf{x}^k)\right|^2}{\sqrt{\mathbf{v}_i^k + \varepsilon}} - \frac{\eta}{4}\sum_{i=1}^d \frac{\left|\mathbf{m}_i^k + \lambda\mathbf{x}_i^k\sqrt{\mathbf{v}_i^k + \varepsilon}\right|^2}{\sqrt{\mathbf{v}_i^k + \varepsilon}} + \frac{\eta}{\sqrt{\varepsilon}}\left\|\nabla f(\mathbf{x}^k) - \mathbf{m}^k\right\|^2\right.
$$

$$
\left. + \eta\sum_{i=1}^d |\lambda\mathbf{x}_i^k|^2\sqrt{\mathbf{v}_i^k + \varepsilon}\Big| \mathcal{F}_{k-1}\right]
$$

$$
\overset{(2)}{\leq} \mathbb{E}_k\left[-\frac{\eta}{2}\sum_{i=1}^d \frac{\left|\nabla_i f(\mathbf{x}^k)\right|^2}{\sqrt{\mathbf{v}_i^k + \varepsilon}} - \frac{\eta}{4}\sum_{i=1}^d \frac{\left|\mathbf{m}_i^k + \lambda\mathbf{x}_i^k\sqrt{\mathbf{v}_i^k + \varepsilon}\right|^2}{\sqrt{\mathbf{v}_i^k + \varepsilon}} + \frac{\eta}{\sqrt{\varepsilon}}\left\|\nabla f(\mathbf{x}^k) - \mathbf{m}^k\right\|^2\right.
$$

$$
\left. + \frac{9\eta\nu}{K^{1/2}}\sum_{i=1}^d \sqrt{\mathbf{v}_i^k + \varepsilon}\Big| \mathcal{F}_{k-1}\right],
$$

*where we let $\eta \leq \frac{\sqrt{\varepsilon}}{2L}$ in $\overset{(1)}{\leq}$ and use Lemma 3 in $\overset{(2)}{\leq}$. Denote*

$$
\widetilde{\mathbf{v}}_i^k = \beta\mathbf{v}_i^{k-1} + (1-\beta)\left(\left|\nabla_i f(\mathbf{x}^k)\right|^2 + \sigma_i^2\right).
$$

*From the concavity of $\sqrt{x}$ and $-\frac{1}{\sqrt{x}}$ and Assumptions 2 and 3, we have*

$$
\mathbb{E}_k\left[\sqrt{\mathbf{v}_i^k + \varepsilon}|\mathcal{F}_{k-1}\right] \leq \sqrt{\mathbb{E}_k\left[\mathbf{v}_i^k|\mathcal{F}_{k-1}\right] + \varepsilon} = \sqrt{\beta\mathbf{v}_i^{k-1} + (1-\beta)\mathbb{E}_k\left[|\mathbf{g}_i^k|^2|\mathcal{F}_{k-1}\right] + \varepsilon}
$$

$$
\leq \sqrt{\beta\mathbf{v}_i^{k-1} + (1-\beta)\left(|\nabla_i f(\mathbf{x}^k)|^2 + \sigma_i^2\right) + \varepsilon} = \sqrt{\widetilde{\mathbf{v}}_i^k + \varepsilon},
$$

$$
-\mathbb{E}_k\left[\frac{\left|\nabla_i f(\mathbf{x}^k)\right|^2}{\sqrt{\mathbf{v}_i^k + \varepsilon}}|\mathcal{F}_{k-1}\right] \leq -\frac{\left|\nabla_i f(\mathbf{x}^k)\right|^2}{\sqrt{\mathbb{E}_k\left[\mathbf{v}_i^k|\mathcal{F}_{k-1}\right] + \varepsilon}} \leq -\frac{\left|\nabla_i f(\mathbf{x}^k)\right|^2}{\sqrt{\widetilde{\mathbf{v}}_i^k + \varepsilon}}.
$$

*Plugging into (13) and rearranging the terms, we have*

$$
\mathbb{E}_k \left[ f(\mathbf{x}^{k+1}) - f^* + \frac{\eta}{4} \sum_{i=1}^{d} \frac{\left| \mathbf{m}_i^k + \lambda \mathbf{x}_i^k \sqrt{\mathbf{v}_i^k + \varepsilon} \right|^2}{\sqrt{\mathbf{v}_i^k + \varepsilon}} - \frac{\eta}{\sqrt{\varepsilon}} \left\| \nabla f(\mathbf{x}^k) - \mathbf{m}^k \right\|^2 \Big| \mathcal{F}_{k-1} \right]
$$

$$
\leq f(\mathbf{x}^k) - f^* - \frac{\eta}{2} \sum_{i=1}^{d} \frac{\left| \nabla_i f(\mathbf{x}^k) \right|^2}{\sqrt{\widetilde{\mathbf{v}}_i^k + \varepsilon}} + \frac{9\eta\nu}{K^{1/2}} \sum_{i=1}^{d} \sqrt{\widetilde{\mathbf{v}}_i^k + \varepsilon}. \tag{14}
$$

*Multiplying both sides of (18) in Lemma 4 by $\frac{\eta}{\sqrt{\varepsilon}(1-\theta)}$ and adding it to (14), we have*

$$
\mathbb{E}_k \left[ f(\mathbf{x}^{k+1}) - f^* + \frac{\eta\theta}{\sqrt{\varepsilon}(1-\theta)} \left\| \nabla f(\mathbf{x}^k) - \mathbf{m}^k \right\|^2 + \frac{\eta}{4} \sum_{i=1}^{d} \frac{\left| \mathbf{m}_i^k + \lambda \mathbf{x}_i^k \sqrt{\mathbf{v}_i^k + \varepsilon} \right|^2}{\sqrt{\mathbf{v}_i^k + \varepsilon}} \Big| \mathcal{F}_{k-1} \right]
$$

$$
\leq f(\mathbf{x}^k) - f^* - \frac{\eta}{2} \sum_{i=1}^{d} \frac{\left| \nabla_i f(\mathbf{x}^k) \right|^2}{\sqrt{\widetilde{\mathbf{v}}_i^k + \varepsilon}} + \frac{9\eta\nu}{K^{1/2}} \sum_{i=1}^{d} \sqrt{\widetilde{\mathbf{v}}_i^k + \varepsilon} + \frac{\eta\theta}{\sqrt{\varepsilon}(1-\theta)} \left\| \nabla f(\mathbf{x}^{k-1}) - \mathbf{m}^{k-1} \right\|^2
$$

$$
+ \frac{L^2 \eta^3}{\varepsilon(1-\theta)^2} \sum_{i=1}^{d} \frac{\left| \mathbf{m}_i^{k-1} + \lambda \mathbf{x}_i^{k-1} \sqrt{\mathbf{v}_i^{k-1} + \varepsilon} \right|^2}{\sqrt{\mathbf{v}_i^{k-1} + \varepsilon}} + \frac{\eta(1-\theta)\sigma_s^2}{\sqrt{\varepsilon}} \tag{15}
$$

$$
\leq f(\mathbf{x}^k) - f^* - \frac{\eta}{2} \sum_{i=1}^{d} \frac{\left| \nabla_i f(\mathbf{x}^k) \right|^2}{\sqrt{\widetilde{\mathbf{v}}_i^k + \varepsilon}} + \frac{9\eta\nu}{K^{1/2}} \sum_{i=1}^{d} \sqrt{\widetilde{\mathbf{v}}_i^k + \varepsilon}
$$

$$
+ \frac{\eta\theta}{\sqrt{\varepsilon}(1-\theta)} \left\| \nabla f(\mathbf{x}^{k-1}) - \mathbf{m}^{k-1} \right\|^2 + \frac{\eta}{4} \sum_{i=1}^{d} \frac{\left| \mathbf{m}_i^{k-1} + \lambda \mathbf{x}_i^{k-1} \sqrt{\mathbf{v}_i^{k-1} + \varepsilon} \right|^2}{\sqrt{\mathbf{v}_i^{k-1} + \varepsilon}} + \frac{\eta(1-\theta)\sigma_s^2}{\sqrt{\varepsilon}},
$$

*where we let $\eta^2 \leq \frac{\varepsilon(1-\theta)^2}{4L^2}$ in the last inequality. For both (14) and (15), taking expectation with respect to $\mathcal{F}_{k-1}$, rearranging the terms, summing (14) with $k = 1$ and (15) over $k = 2, 3, \cdots, K$, we have*

$$
\mathbb{E}_{\mathcal{F}_K} \left[ f(\mathbf{x}^{K+1}) - f^* + \frac{\eta\theta}{\sqrt{\varepsilon}(1-\theta)} \left\| \nabla f(\mathbf{x}^K) - \mathbf{m}^K \right\|^2 + \frac{\eta}{4} \sum_{i=1}^{d} \frac{\left| \mathbf{m}_i^K + \lambda \mathbf{x}_i^K \sqrt{\mathbf{v}_i^K + \varepsilon} \right|^2}{\sqrt{\mathbf{v}_i^K + \varepsilon}} \right]
$$

$$
\leq f(\mathbf{x}^1) - f^* + \sum_{k=1}^{K} \sum_{i=1}^{d} \mathbb{E}_{\mathcal{F}_{k-1}} \left[ -\frac{\eta}{2} \frac{\left| \nabla_i f(\mathbf{x}^k) \right|^2}{\sqrt{\widetilde{\mathbf{v}}_i^k + \varepsilon}} + \frac{9\eta\nu}{K^{1/2}} \sqrt{\widetilde{\mathbf{v}}_i^k + \varepsilon} \right]
$$

$$
+ \frac{\eta\theta}{\sqrt{\varepsilon}(1-\theta)} \mathbb{E}_{\mathcal{F}_1} \left[ \|\nabla f(\mathbf{x}^1) - \mathbf{m}^1\|^2 \right] + \frac{\eta}{\sqrt{\varepsilon}} \mathbb{E}_{\mathcal{F}_1} \left[ \|\nabla f(\mathbf{x}^1) - \mathbf{m}^1\|^2 \right] + \frac{(K-1)\eta(1-\theta)\sigma_s^2}{\sqrt{\varepsilon}}
$$

$$
\overset{(3)}{\leq} f(\mathbf{x}^1) - f^* - \frac{\eta}{2} \sum_{k=1}^{K} \sum_{i=1}^{d} \mathbb{E}_{\mathcal{F}_{k-1}} \left[ \frac{\left| \nabla_i f(\mathbf{x}^k) \right|^2}{\sqrt{\widetilde{\mathbf{v}}_i^k + \varepsilon}} \right]
$$

$$
+ \frac{9\eta\nu}{K^{1/2}} \left( K\|\boldsymbol{\sigma}\|_1 + Kd\sqrt{\varepsilon} + 2 \sum_{k=1}^{K} \sum_{i=1}^{d} \mathbb{E}_{\mathcal{F}_{k-1}} \left[ \frac{\left| \nabla_i f(\mathbf{x}^k) \right|^2}{\sqrt{\widetilde{\mathbf{v}}_i^k + \varepsilon}} \right] \right) \tag{16}
$$

$$
+ \frac{\eta}{\sqrt{\varepsilon}(1-\theta)} \mathbb{E}_{\mathcal{F}_1} \left[ \|\nabla f(\mathbf{x}^1) - \mathbf{m}^1\|^2 \right] + \frac{(K-1)\eta(1-\theta)\sigma_s^2}{\sqrt{\varepsilon}}
$$

$$
\overset{(4)}{\leq} f(\mathbf{x}^1) - f^* - \frac{\eta}{4} \sum_{k=1}^{K} \sum_{i=1}^{d} \mathbb{E}_{\mathcal{F}_{k-1}} \left[ \frac{\left| \nabla_i f(\mathbf{x}^k) \right|^2}{\sqrt{\widetilde{\mathbf{v}}_i^k + \varepsilon}} \right]
$$

$$
+ 18\eta\nu d\sqrt{K\varepsilon} + \frac{\eta}{\sqrt{\varepsilon}(1-\theta)} \mathbb{E}_{\mathcal{F}_1} \left[ \|\nabla f(\mathbf{x}^1) - \mathbf{m}^1\|^2 \right] + \frac{(K-1)\eta(1-\theta)\sigma_s^2}{\sqrt{\varepsilon}}
$$

$$
\overset{(5)}{\leq} f(\mathbf{x}^1) - f^* - \frac{\eta}{4} \sum_{k=1}^{K} \sum_{i=1}^{d} \mathbb{E}_{\mathcal{F}_{k-1}} \left[ \frac{\left| \nabla_i f(\mathbf{x}^k) \right|^2}{\sqrt{\widetilde{\mathbf{v}}_i^k + \varepsilon}} \right]
$$

$$
+ 18\eta\nu d\sqrt{K\varepsilon} + \frac{2\eta L(f(\mathbf{x}^1) - f^*)}{\sqrt{\varepsilon}(1-\theta)} + \frac{\eta(1-\theta)\sigma_s^2}{\sqrt{\varepsilon}} + + \frac{(K-1)\eta(1-\theta)\sigma_s^2}{\sqrt{\varepsilon}},
$$

*where we use Lemma 5 in $\overset{(3)}{\leq}$, let $\frac{9\nu}{K^{1/2}} \leq \frac{1}{8}$ and $\varepsilon \geq \frac{\sigma_s^2}{d}$ such that $\|\boldsymbol{\sigma}\|_1 \leq \sqrt{d}\|\boldsymbol{\sigma}\|_2 = \sqrt{d}\sigma_s \leq d\sqrt{\varepsilon}$ in $\overset{(4)}{\leq}$, and use $\mathbf{m}^0 = 0$,*

$$f^* \leq f\left(\mathbf{x} - \frac{1}{L}\nabla f(\mathbf{x})\right) \leq f(\mathbf{x}) - \frac{1}{L}\langle\nabla f(\mathbf{x}), \nabla f(\mathbf{x})\rangle + \frac{L}{2}\left\|\frac{1}{L}\nabla f(\mathbf{x})\right\|^2 = f(\mathbf{x}) - \frac{1}{2L}\|\nabla f(\mathbf{x})\|^2,$$

*and*

$$\begin{aligned}
\mathbb{E}_{\mathcal{F}_1}\left[\|\nabla f(\mathbf{x}^1) - \mathbf{m}^1\|^2\right] &= \mathbb{E}_{\mathcal{F}_1}\left[\|\theta\nabla f(\mathbf{x}^1) + (1-\theta)(\nabla f(\mathbf{x}^1) - \mathbf{g}^1)\|^2\right] \\
&= \theta^2\|\nabla f(\mathbf{x}^1)\|^2 + (1-\theta)^2\mathbb{E}_{\mathcal{F}_1}\left[\|\nabla f(\mathbf{x}^1) - \mathbf{g}^1\|^2\right] \\
&\leq 2L(f(\mathbf{x}^1) - f^*) + (1-\theta)^2\sigma_s^2
\end{aligned}$$

*in $\overset{(5)}{\leq}$. So from (16), we have*

$$\begin{aligned}
&\sum_{k=1}^K \sum_{i=1}^d \mathbb{E}_{\mathcal{F}_{k-1}}\left[\frac{|\nabla_i f(\mathbf{x}^k)|^2}{\sqrt{\widetilde{\mathbf{v}}_i^k + \varepsilon}}\right] \\
&\leq \frac{4(f(\mathbf{x}^1) - f^*)}{\eta} + 72\nu d\sqrt{K\varepsilon} + \frac{8L(f(\mathbf{x}^1) - f^*)}{\sqrt{\varepsilon}(1-\theta)} + \frac{4K(1-\theta)\sigma_s^2}{\sqrt{\varepsilon}} \\
&\leq \frac{4(f(\mathbf{x}^1) - f^*)}{\eta} + 72\nu d\sqrt{K\varepsilon} + \frac{8L(f(\mathbf{x}^1) - f^*)}{\sqrt{\varepsilon}(1-\theta)} + \frac{4K(1-\theta)\hat{\sigma}_s^2}{\sqrt{\varepsilon}},
\end{aligned} \tag{17}$$

*where we denote $\hat{\sigma}_s^2 = \max\left\{\sigma_s^2, \frac{L(f(\mathbf{x}^1)-f^*)}{K\gamma^2}\right\}$ with any constant $\gamma \in (0, 1]$.*

*Recall that we require the parameters satisfying the following relations in the above proof*

$$\eta \leq \frac{\sqrt{\varepsilon}}{2L}, \quad \eta^2 \leq \frac{\varepsilon(1-\theta)^2}{4L^2}, \quad \frac{9\nu}{K^{1/2}} \leq \frac{1}{8}, \quad \varepsilon \geq \frac{\sigma_s^2}{d}$$

*and*

$$\eta\lambda \leq \frac{\sqrt{\nu}}{2K^{5/4}}, \quad \frac{\sqrt{\nu}}{K^{1/4}} < 1, \quad \|\mathbf{x}^1\|_\infty \leq \frac{\sqrt{\nu}}{K^{1/4}\lambda}, \quad \theta \leq \beta \leq \sqrt{\theta} < 1$$

*in Lemma 3.*

*Recalling the definition of $\hat{\sigma}_s$ and letting $\varepsilon = \frac{\hat{\sigma}_s^2}{d}$, $1 - \theta = \sqrt{\frac{L(f(\mathbf{x}^1)-f^*)}{K\hat{\sigma}_s^2}}$, $\eta = \sqrt{\frac{\varepsilon(f(\mathbf{x}^1)-f^*)}{4K\hat{\sigma}_s^2 L}} = \sqrt{\frac{f(\mathbf{x}^1)-f^*}{4KdL}}$, $\nu = \frac{1}{72d}\sqrt{\frac{\hat{\sigma}_s^2 L(f(\mathbf{x}^1)-f^*)}{\varepsilon^2}} = \frac{1}{72}\sqrt{\frac{L(f(\mathbf{x}^1)-f^*)}{\hat{\sigma}_s^2}}$, $\lambda \leq \frac{\sqrt{\nu}}{2K^{5/4}\eta} = \frac{\sqrt{d}}{\sqrt{72}K^{3/4}}\sqrt[4]{\frac{L^3}{\hat{\sigma}_s^2(f(\mathbf{x}^1)-f^*)}}$, and $\|\mathbf{x}^1\|_\infty \leq \sqrt{\frac{K(f(\mathbf{x}^1)-f^*)}{dL}} = 2K\eta \leq \frac{\sqrt{\nu}}{K^{1/4}\lambda}$, the above requirements are satisfied. So we have from (17) that*

$$\sum_{k=1}^K \sum_{i=1}^d \mathbb{E}_{\mathcal{F}_{k-1}}\left[\frac{|\nabla_i f(\mathbf{x}^k)|^2}{\sqrt{\widetilde{\mathbf{v}}_i^k + \varepsilon}}\right] \leq 21\sqrt{\frac{K\hat{\sigma}_s^2 L(f(\mathbf{x}^1)-f^*)}{\varepsilon}}.$$

*Using Holder's inequality and Lemma 5, we have*

$$\begin{aligned}
&\left(\sum_{k=1}^K \mathbb{E}_{\mathcal{F}_{k-1}}\left[\|\nabla f(\mathbf{x}^k)\|_1\right]\right)^2 \\
&\leq \left(\sum_{k=1}^K \sum_{i=1}^d \mathbb{E}_{\mathcal{F}_{k-1}}\left[\frac{|\nabla_i f(\mathbf{x}^k)|^2}{\sqrt{\widetilde{\mathbf{v}}_i^k + \varepsilon}}\right]\right)\left(\sum_{k=1}^K \sum_{i=1}^d \mathbb{E}_{\mathcal{F}_{k-1}}\left[\sqrt{\widetilde{\mathbf{v}}_i^k + \varepsilon}\right]\right) \\
&\leq \left(\sum_{k=1}^K \sum_{i=1}^d \mathbb{E}_{\mathcal{F}_{k-1}}\left[\frac{|\nabla_i f(\mathbf{x}^k)|^2}{\sqrt{\widetilde{\mathbf{v}}_i^k + \varepsilon}}\right]\right)\left(K\|\boldsymbol{\sigma}\|_1 + Kd\sqrt{\varepsilon} + 2\sum_{k=1}^K \sum_{i=1}^d \mathbb{E}_{\mathcal{F}_{k-1}}\left[\frac{|\nabla_i f(\mathbf{x}^k)|^2}{\sqrt{\widetilde{\mathbf{v}}_i^k + \varepsilon}}\right]\right) \\
&\leq \left(21\sqrt{\frac{K\hat{\sigma}_s^2 L(f(\mathbf{x}^1)-f^*)}{\varepsilon}}\right)\left(42\sqrt{\frac{K\hat{\sigma}_s^2 L(f(\mathbf{x}^1)-f^*)}{\varepsilon}} + K\|\boldsymbol{\sigma}\|_1 + Kd\sqrt{\varepsilon}\right)
\end{aligned}$$

*and*

$$\begin{aligned}
&\frac{1}{K}\sum_{k=1}^K \mathbb{E}_{\mathcal{F}_{k-1}}\left[\|\nabla f(\mathbf{x}^k)\|_1\right] \\
&\leq \frac{1}{K}\left(30\sqrt{\frac{K\hat{\sigma}_s^2 L(f(\mathbf{x}^1)-f^*)}{\varepsilon}} + 5\sqrt[4]{\frac{K\hat{\sigma}_s^2 L(f(\mathbf{x}^1)-f^*)}{\varepsilon}\left(K\|\boldsymbol{\sigma}\|_1 + Kd\sqrt{\varepsilon}\right)^2}\right) \\
&\leq 30\sqrt{\frac{dL(f(\mathbf{x}^1)-f^*)}{K}} + \frac{8\sqrt{d}}{K^{1/4}}\sqrt[4]{\hat{\sigma}_s^2 L(f(\mathbf{x}^1)-f^*)}
\end{aligned}$$

by letting $\varepsilon = \frac{\hat{\sigma}_s^2}{d}$ and using $\|\boldsymbol{\sigma}\|_1 \leq \sqrt{d}\|\boldsymbol{\sigma}\|_2 = \sqrt{d}\sigma_s \leq d\sqrt{\varepsilon}$. At last, from Lemma 3 and the settings of $\nu$ and $\hat{\sigma}_s$, we have

$$\lambda\|\mathbf{x}^k\|_\infty \leq \frac{3\sqrt{\nu}}{K^{1/4}} = \frac{3}{\sqrt{72}} \sqrt[4]{\frac{L(f(\mathbf{x}^1)-f^*)}{K\hat{\sigma}_s^2}} \leq \frac{3}{\sqrt{72}}$$

for all $k = 1, 2, \cdots, K$, leading to $\|\mathbf{x}^k\|_\infty < \frac{1}{\lambda}$.

# B Proof of Corollary 1

We give the complete description of Corollary 1 in the following corollary.

**Corollary 2** *Suppose that Assumptions 1-3 hold. Define $\hat{\sigma}_s^2 = \max\left\{\sigma_s^2, \frac{L(f(\mathbf{x}^1)-f^*)}{K\gamma^2}\right\}$ with any constant $\gamma \in (0, 1]$. Let $1 - \theta = \sqrt{\frac{L(f(\mathbf{x}^1)-f^*)}{K\hat{\sigma}_s^2}}$, $0 \leq \beta \leq 1$, $\eta = \sqrt{\frac{f(\mathbf{x}^1)-f^*}{4dKL}}$, and $\varepsilon = \frac{\hat{\sigma}_s^2}{d}$. Then for Adam, we have*

$$\frac{1}{K}\sum_{k=1}^K \mathbb{E}\left[\|\nabla f(\mathbf{x}^k)\|_1\right] \leq \frac{6\sqrt{d}}{K^{1/4}} \sqrt[4]{\hat{\sigma}_s^2 L(f(\mathbf{x}^1)-f^*)} + 15\sqrt{\frac{dL(f(\mathbf{x}^1)-f^*)}{K}}.$$

*Specially, when $\sigma_s^2 \leq \frac{L(f(\mathbf{x}^1)-f^*)}{K\gamma^2}$, we have $1 - \theta = \gamma$, $0 \leq \beta \leq 1$, $\eta = \sqrt{\frac{f(\mathbf{x}^1)-f^*}{4KdL}}$, $\varepsilon = \frac{L(f(\mathbf{x}^1)-f^*)}{dK\gamma^2}$, and accordlingly*

$$\frac{1}{K}\sum_{k=1}^K \mathbb{E}\left[\|\nabla f(\mathbf{x}^k)\|_1\right] \leq 21\sqrt{\frac{dL(f(\mathbf{x}^1)-f^*)}{K\gamma}}.$$

**Proof 2** *When $\lambda = 0$, the $\frac{9\eta\nu}{K^{1/2}}\sum_{i=1}^d \sqrt{\mathbf{v}_i^k + \varepsilon}$ term disappears in (13) in the proof of Theorem 1, and (16) becomes*

$$\mathbb{E}_{\mathcal{F}_K}\left[f(\mathbf{x}^{K+1}) - f^* + \frac{\eta\theta}{\sqrt{\varepsilon}(1-\theta)}\left\|\nabla f(\mathbf{x}^K) - \mathbf{m}^K\right\|^2 + \frac{\eta}{4}\sum_{i=1}^d \frac{\left|\mathbf{m}_i^K + \lambda\mathbf{x}_i^K\sqrt{\mathbf{v}_i^K + \varepsilon}\right|^2}{\sqrt{\mathbf{v}_i^K + \varepsilon}}\right]$$

$$\leq f(\mathbf{x}^1) - f^* - \frac{\eta}{2}\sum_{k=1}^K\sum_{i=1}^d \mathbb{E}_{\mathcal{F}_{k-1}}\left[\frac{|\nabla_i f(\mathbf{x}^k)|^2}{\sqrt{\widetilde{\mathbf{v}}_i^k + \varepsilon}}\right] + \frac{2\eta L(f(\mathbf{x}^1)-f^*)}{\sqrt{\varepsilon}(1-\theta)} + \frac{K\eta(1-\theta)\sigma_s^2}{\sqrt{\varepsilon}},$$

*where the term $18\eta\nu d\sqrt{K\varepsilon}$ disappears because we do not need Lemma 5 to bound $\frac{9\eta\nu}{K^{1/2}}\sum_{k=1}^K\sum_{i=1}^d \mathbb{E}_{\mathcal{F}_{k-1}}\left[\sqrt{\widetilde{\mathbf{v}}_i^k + \varepsilon}\right]$ any more.*

*Similar to the proof of Theorem 1, we have*

$$\sum_{k=1}^K\sum_{i=1}^d \mathbb{E}_{\mathcal{F}_{k-1}}\left[\frac{|\nabla_i f(\mathbf{x}^k)|^2}{\sqrt{\widetilde{\mathbf{v}}_i^k + \varepsilon}}\right] \leq \frac{2(f(\mathbf{x}^1)-f^*)}{\eta} + \frac{4L(f(\mathbf{x}^1)-f^*)}{\sqrt{\varepsilon}(1-\theta)} + \frac{2K(1-\theta)\sigma_s^2}{\sqrt{\varepsilon}}$$

$$\leq 10\sqrt{\frac{K\hat{\sigma}_s^2 L(f(\mathbf{x}^1)-f^*)}{\varepsilon}}.$$

*Comparing with (17), we see that the term $72\nu d\sqrt{K\varepsilon}$ disappears. Following the proof of Theorem 1, we have the conclusion. Note that we do not use Lemmas 2 and 3 in the proof of Corollary 1, so Corollary 1 does not require $\theta \leq \beta \leq \sqrt{\theta}$ and $\|\mathbf{x}^1\|_\infty \leq \frac{\sqrt{\nu}}{K^{1/4}\lambda}$ any more.*

# C Supporting Lemmas

**Lemma 2** *Suppose $\mathbf{m}^0 = 0$, $\mathbf{v}^0 = 0$, and $\theta \leq \beta \leq \sqrt{\theta} < 1$, then we have*

$$\frac{|\mathbf{m}_i^k|^2}{\mathbf{v}_i^k} \leq \frac{(1-\theta)^2\beta}{(1-\beta)(\beta-\theta^2)} \leq 4.$$

**Proof 3** *From the recursions of $\mathbf{m}_i^k$ and $\mathbf{v}_i^k$, we have*

$$\mathbf{m}_i^k = \theta^k\mathbf{m}_i^0 + (1-\theta)\sum_{r=1}^k \theta^{k-r}\mathbf{g}_i^r = (1-\theta)\sum_{r=1}^k \theta^{k-r}\mathbf{g}_i^r,$$

$$\mathbf{v}_i^k = \beta^k\mathbf{v}_i^0 + (1-\beta)\sum_{r=1}^k \beta^{k-r}|\mathbf{g}_i^r|^2 = (1-\beta)\sum_{r=1}^k \beta^{k-r}|\mathbf{g}_i^r|^2.$$

*Using Holder's inequality, we have*

$$|\mathbf{m}_i^k|^2 = (1-\theta)^2 \left( \sum_{r=1}^k \theta^{k-r} \mathbf{g}_i^r \right)^2 \le (1-\theta)^2 \left( \sum_{r=1}^k \beta^{k-r} |\mathbf{g}_i^r|^2 \right) \left( \sum_{r=1}^k \left( \frac{\theta^2}{\beta} \right)^{k-r} \right)$$

$$= \mathbf{v}_i^k \frac{(1-\theta)^2}{1-\beta} \sum_{r=1}^k \left( \frac{\theta^2}{\beta} \right)^{k-r} \le \mathbf{v}_i^k \frac{(1-\theta)^2}{1-\beta} \frac{1}{1-\frac{\theta^2}{\beta}} \overset{(1)}{\le} \mathbf{v}_i^k \frac{(1-\theta)^2}{(1-\beta)^2}$$

$$\overset{(2)}{\le} \mathbf{v}_i^k \frac{(1-\sqrt{\theta})^2(1+\sqrt{\theta})^2}{(1-\sqrt{\theta})^2} \le \mathbf{v}_i^k (1+\sqrt{\theta})^2 \le 4\mathbf{v}_i^k,$$

*where we use $\theta \le \beta$ in $\overset{(1)}{\le}$ and $\beta \le \sqrt{\theta}$ in $\overset{(2)}{\le}$.*

**Lemma 3** *Suppose $\eta\lambda \le \frac{\sqrt{\nu}}{2K^{5/4}}$, $\|\mathbf{x}^1\|_\infty \le \frac{\sqrt{\nu}}{K^{1/4}\lambda}$, $\frac{\sqrt{\nu}}{K^{1/4}} < 1$, and $\theta \le \beta \le \sqrt{\theta} < 1$, then we have*

$$\lambda \|\mathbf{x}^k\|_\infty \le \frac{3\sqrt{\nu}}{K^{1/4}}, \quad \forall k = 1, 2, \cdots, K.$$

**Proof 4** *From the update of $\mathbf{x}^{k+1}$, we have*

$$\|\mathbf{x}^{k+1}\|_\infty - \frac{2}{\lambda} = \left\| (1-\eta\lambda)\mathbf{x}^k - \frac{\eta}{\sqrt{\mathbf{v}^k}+\varepsilon} \odot \mathbf{m}^k \right\|_\infty - \frac{2}{\lambda}$$

$$\le (1-\eta\lambda)\|\mathbf{x}^k\|_\infty + \left\| \frac{\eta}{\sqrt{\mathbf{v}^k}+\varepsilon} \odot \mathbf{m}^k \right\|_\infty - \frac{2}{\lambda}$$

$$\overset{(1)}{\le} (1-\eta\lambda)\|\mathbf{x}^k\|_\infty + 2\eta - \frac{2}{\lambda}$$

$$= (1-\eta\lambda)\left( \|\mathbf{x}^k\|_\infty - \frac{2}{\lambda} \right)$$

$$\le (1-\eta\lambda)^k \left( \|\mathbf{x}^1\|_\infty - \frac{2}{\lambda} \right)$$

$$\le -\frac{1}{\lambda}(1-\eta\lambda)^k \left( 2 - \frac{\sqrt{\nu}}{K^{1/4}} \right),$$

*where we use Lemma 2 in $\overset{(1)}{\le}$. Since $\ln x \le x - 1$ and $e^x \ge x + 1$ for any $x > 0$ and $\eta\lambda \le \frac{\sqrt{\nu}}{2K^{5/4}} \le \frac{1}{2}$, we have for any $k \le K$ that*

$$k\ln(1-\eta\lambda) = -k\ln\frac{1}{1-\eta\lambda} \ge -K\left( \frac{1}{1-\eta\lambda} - 1 \right) = -\frac{K\eta\lambda}{1-\eta\lambda} \ge -\frac{\sqrt{\nu}}{K^{1/4}},$$

$$(1-\eta\lambda)^k \ge e^{-\frac{\sqrt{\nu}}{K^{1/4}}} \ge 1 - \frac{\sqrt{\nu}}{K^{1/4}},$$

*and*

$$\|\mathbf{x}^{k+1}\|_\infty - \frac{2}{\lambda} \le -\frac{1}{\lambda}\left( 1 - \frac{\sqrt{\nu}}{K^{1/4}} \right)\left( 2 - \frac{\sqrt{\nu}}{K^{1/4}} \right) \le -\frac{2}{\lambda} + \frac{3}{\lambda}\frac{\sqrt{\nu}}{K^{1/4}}.$$

**Lemma 4** *Suppose that Assumptions 1-3 hold. Then we have*

$$\mathbb{E}_k\left[ \left\| \mathbf{m}^k - \nabla f(\mathbf{x}^k) \right\|^2 | \mathcal{F}_{k-1} \right]$$

$$\le \theta \left\| \mathbf{m}^{k-1} - \nabla f(\mathbf{x}^{k-1}) \right\|^2 + \frac{L^2\eta^2}{\sqrt{\varepsilon}(1-\theta)} \sum_{i=1}^d \frac{\left| \mathbf{m}_i^{k-1} + \lambda\mathbf{x}_i^{k-1}\sqrt{\mathbf{v}_i^{k-1}+\varepsilon} \right|^2}{\sqrt{\mathbf{v}_i^{k-1}+\varepsilon}} + (1-\theta)^2\sigma_s^2. \tag{18}$$

**Proof 5** *Denoting $\zeta^k = \mathbf{g}^k - \nabla f(\mathbf{x}^k)$, from the update of $\mathbf{m}^k$, we have*

$$\mathbf{m}^k - \nabla f(\mathbf{x}^k) = \theta\mathbf{m}^{k-1} + (1-\theta)\mathbf{g}^k - \nabla f(\mathbf{x}^k)$$

$$= \theta\left( \mathbf{m}^{k-1} - \nabla f(\mathbf{x}^{k-1}) \right) + (1-\theta)\left( \nabla f(\mathbf{x}^k) + \zeta^k \right) - \nabla f(\mathbf{x}^k) + \theta\nabla f(\mathbf{x}^{k-1})$$

$$= \theta\left( \mathbf{m}^{k-1} - \nabla f(\mathbf{x}^{k-1}) \right) + (1-\theta)\zeta^k - \theta\left( \nabla f(\mathbf{x}^k) - \nabla f(\mathbf{x}^{k-1}) \right)$$

*and*

$$\mathbb{E}_k\left[\|\mathbf{m}^k - \nabla f(\mathbf{x}^k)\|^2|\mathcal{F}_{k-1}\right]$$

$$\leq \left\|\theta\left(\mathbf{m}^{k-1} - \nabla f(\mathbf{x}^{k-1})\right) - \theta\left(\nabla f(\mathbf{x}^k) - \nabla f(\mathbf{x}^{k-1})\right)\right\|^2 + (1-\theta)^2\sigma_s^2$$

$$\leq \theta^2\left(\left(1 + \frac{1-\theta}{\theta}\right)\left\|\mathbf{m}^{k-1} - \nabla f(\mathbf{x}^{k-1})\right\|^2 + \left(1 + \frac{\theta}{1-\theta}\right)\left\|\nabla f(\mathbf{x}^k) - \nabla f(\mathbf{x}^{k-1})\right\|^2\right) + (1-\theta)^2\sigma_s^2$$

$$\leq \theta\left\|\mathbf{m}^{k-1} - \nabla f(\mathbf{x}^{k-1})\right\|^2 + \frac{L^2}{1-\theta}\left\|\mathbf{x}^k - \mathbf{x}^{k-1}\right\|^2 + (1-\theta)^2\sigma_s^2$$

$$= \theta\left\|\mathbf{m}^{k-1} - \nabla f(\mathbf{x}^{k-1})\right\|^2 + \frac{L^2\eta^2}{1-\theta}\sum_{i=1}^d \frac{\left|\mathbf{m}_i^{k-1} + \lambda\mathbf{x}_i^{k-1}\sqrt{\mathbf{v}_i^{k-1} + \varepsilon}\right|^2}{\mathbf{v}_i^{k-1} + \varepsilon} + (1-\theta)^2\sigma_s^2$$

$$\leq \theta\left\|\mathbf{m}^{k-1} - \nabla f(\mathbf{x}^{k-1})\right\|^2 + \frac{L^2\eta^2}{\sqrt{\varepsilon}(1-\theta)}\sum_{i=1}^d \frac{\left|\mathbf{m}_i^{k-1} + \lambda\mathbf{x}_i^{k-1}\sqrt{\mathbf{v}_i^{k-1} + \varepsilon}\right|^2}{\sqrt{\mathbf{v}_i^{k-1} + \varepsilon}} + (1-\theta)^2\sigma_s^2.$$

The following lemma is modified from [10]. We give the proof here only for the sake of completeness.

**Lemma 5** *Suppose that Assumptions 1-3 hold. Let $\beta \leq 1$ and $\mathbf{v}^0 = 0$. Then we have*

$$\sum_{k=1}^K\sum_{i=1}^d \mathbb{E}_{\mathcal{F}_{k-1}}\left[\sqrt{\widetilde{\mathbf{v}}_i^k + \varepsilon}\right] \leq K\|\boldsymbol{\sigma}\|_1 + Kd\sqrt{\varepsilon} + 2\sum_{t=1}^K\sum_{i=1}^d \mathbb{E}_{\mathcal{F}_{t-1}}\left[\frac{|\nabla_i f(\mathbf{x}^t)|^2}{\sqrt{\mathbf{v}_i^t + \varepsilon}}\right].$$

**Proof 6** *From the definition of $\widetilde{\mathbf{v}}_i^k$, we have*

$$\mathbb{E}_{\mathcal{F}_{k-1}}\left[\sqrt{\widetilde{\mathbf{v}}_i^k + \varepsilon}\right]$$

$$= \mathbb{E}_{\mathcal{F}_{k-1}}\left[\sqrt{\beta\mathbf{v}_i^{k-1} + (1-\beta)\left(|\nabla_i f(\mathbf{x}^k)|^2 + \sigma_i^2\right) + \varepsilon}\right]$$

$$= \mathbb{E}_{\mathcal{F}_{k-1}}\left[\frac{\beta\mathbf{v}_i^{k-1} + (1-\beta)\sigma_i^2 + \varepsilon}{\sqrt{\beta\mathbf{v}_i^{k-1} + (1-\beta)\left(|\nabla_i f(\mathbf{x}^k)|^2 + \sigma_i^2\right) + \varepsilon}} + \frac{(1-\beta)\left|\nabla_i f(\mathbf{x}^k)\right|^2}{\sqrt{\beta\mathbf{v}_i^{k-1} + (1-\beta)\left(|\nabla_i f(\mathbf{x}^k)|^2 + \sigma_i^2\right) + \varepsilon}}\right]$$

$$\leq \mathbb{E}_{\mathcal{F}_{k-1}}\left[\sqrt{\beta\mathbf{v}_i^{k-1} + (1-\beta)\sigma_i^2 + \varepsilon}\right] + (1-\beta)\mathbb{E}_{\mathcal{F}_{k-1}}\left[\frac{|\nabla_i f(\mathbf{x}^k)|^2}{\sqrt{\widetilde{\mathbf{v}}_i^k + \varepsilon}}\right].$$

*Consider the first part in the general case. From the recursion of $\mathbf{v}_i^k$, we have*

$$\mathbb{E}_{\mathcal{F}_{k-t}}\left[\sqrt{\beta^t\mathbf{v}_i^{k-t} + (1-\beta^t)\sigma_i^2 + \varepsilon}\right]$$

$$= \mathbb{E}_{\mathcal{F}_{k-t}}\left[\sqrt{\beta^{t+1}\mathbf{v}_i^{k-t-1} + \beta^t(1-\beta)|\mathbf{g}_i^{k-t}|^2 + (1-\beta^t)\sigma_i^2 + \varepsilon}\right]$$

$$= \mathbb{E}_{\mathcal{F}_{k-t-1}}\left[\mathbb{E}_{k-t}\left[\sqrt{\beta^{t+1}\mathbf{v}_i^{k-t-1} + \beta^t(1-\beta)|\mathbf{g}_i^{k-t}|^2 + (1-\beta^t)\sigma_i^2 + \varepsilon}\Big|\mathcal{F}_{k-t-1}\right]\right]$$

$$\overset{(1)}{\leq} \mathbb{E}_{\mathcal{F}_{k-t-1}}\left[\sqrt{\beta^{t+1}\mathbf{v}_i^{k-t-1} + \beta^t(1-\beta)\mathbb{E}_{k-t}\left[|\mathbf{g}_i^{k-t}|^2|\mathcal{F}_{k-t-1}\right] + (1-\beta^t)\sigma_i^2 + \varepsilon}\right]$$

$$\overset{(2)}{\leq} \mathbb{E}_{\mathcal{F}_{k-t-1}}\left[\sqrt{\beta^{t+1}\mathbf{v}_i^{k-t-1} + \beta^t(1-\beta)\left(|\nabla_i f(\mathbf{x}^{k-t})|^2 + \sigma_i^2\right) + (1-\beta^t)\sigma_i^2 + \varepsilon}\right]$$

$$= \mathbb{E}_{\mathcal{F}_{k-t-1}}\left[\sqrt{\beta^{t+1}\mathbf{v}_i^{k-t-1} + \beta^t(1-\beta)|\nabla_i f(\mathbf{x}^{k-t})|^2 + (1-\beta^{t+1})\sigma_i^2 + \varepsilon}\right]$$

$$= \mathbb{E}_{\mathcal{F}_{k-t-1}}\left[\frac{\beta^{t+1}\mathbf{v}_i^{k-t-1} + (1-\beta^{t+1})\sigma_i^2 + \varepsilon}{\sqrt{\beta^{t+1}\mathbf{v}_i^{k-t-1} + \beta^t(1-\beta)|\nabla_i f(\mathbf{x}^{k-t})|^2 + (1-\beta^{t+1})\sigma_i^2 + \varepsilon}}\right]$$

$$+ \mathbb{E}_{\mathcal{F}_{k-t-1}}\left[\frac{\beta^t(1-\beta)|\nabla_i f(\mathbf{x}^{k-t})|^2}{\sqrt{\beta^{t+1}\mathbf{v}_i^{k-t-1} + \beta^t(1-\beta)|\nabla_i f(\mathbf{x}^{k-t})|^2 + (1-\beta^{t+1})\sigma_i^2 + \varepsilon}}\right]$$

$$\leq \mathbb{E}_{\mathcal{F}_{k-t-1}} \left[ \sqrt{\beta^{t+1}\mathbf{v}_i^{k-t-1} + (1-\beta^{t+1})\sigma_i^2 + \varepsilon} \right]$$

$$+ \mathbb{E}_{\mathcal{F}_{k-t-1}} \left[ \frac{\beta^t(1-\beta)|\nabla_i f(\mathbf{x}^{k-t})|^2}{\sqrt{\beta^{t+1}\mathbf{v}_i^{k-t-1} + \beta^t(1-\beta)|\nabla_i f(\mathbf{x}^{k-t})|^2 + (\beta^t - \beta^{t+1})\sigma_i^2 + \beta^t\varepsilon}} \right]$$

$$= \mathbb{E}_{\mathcal{F}_{k-t-1}} \left[ \sqrt{\beta^{t+1}\mathbf{v}_i^{k-t-1} + (1-\beta^{t+1})\sigma_i^2 + \varepsilon} \right] + \sqrt{\beta^t}(1-\beta)\mathbb{E}_{\mathcal{F}_{k-t-1}} \left[ \frac{|\nabla_i f(\mathbf{x}^{k-t})|^2}{\sqrt{\widetilde{\mathbf{v}}_i^{k-t} + \varepsilon}} \right],$$

*where we use the concavity of $\sqrt{x}$ in $\overset{(1)}{\leq}$ and Assumptions 2 and 3 in $\overset{(2)}{\leq}$. Applying the above inequality recursively for $t = 1, 2, \cdots, k-1$, we have*

$$\mathbb{E}_{\mathcal{F}_{k-1}} \left[ \sqrt{\beta\mathbf{v}_i^{k-1} + (1-\beta)\sigma_i^2 + \varepsilon} \right]$$

$$\leq \sqrt{\beta^k\mathbf{v}_i^0 + (1-\beta^k)\sigma_i^2 + \varepsilon} + \sum_{t=1}^{k-1} \sqrt{\beta^{k-t}}(1-\beta)\mathbb{E}_{\mathcal{F}_{t-1}} \left[ \frac{|\nabla_i f(\mathbf{x}^t)|^2}{\sqrt{\widetilde{\mathbf{v}}_i^t + \varepsilon}} \right]$$

*and*

$$\mathbb{E}_{\mathcal{F}_{k-1}} \left[ \sqrt{\widetilde{\mathbf{v}}_i^k + \varepsilon} \right] \leq \sqrt{\beta^k\mathbf{v}_i^0 + (1-\beta^k)\sigma_i^2 + \varepsilon} + \sum_{t=1}^{k} \sqrt{\beta^{k-t}}(1-\beta)\mathbb{E}_{\mathcal{F}_{t-1}} \left[ \frac{|\nabla_i f(\mathbf{x}^t)|^2}{\sqrt{\widetilde{\mathbf{v}}_i^t + \varepsilon}} \right]$$

$$\leq \sqrt{\sigma_i^2 + \varepsilon} + \sum_{t=1}^{k} \sqrt{\beta^{k-t}}(1-\beta)\mathbb{E}_{\mathcal{F}_{t-1}} \left[ \frac{|\nabla_i f(\mathbf{x}^t)|^2}{\sqrt{\widetilde{\mathbf{v}}_i^t + \varepsilon}} \right]$$

$$\leq \sigma_i + \sqrt{\varepsilon} + \sum_{t=1}^{k} \sqrt{\beta^{k-t}}(1-\beta)\mathbb{E}_{\mathcal{F}_{t-1}} \left[ \frac{|\nabla_i f(\mathbf{x}^t)|^2}{\sqrt{\widetilde{\mathbf{v}}_i^t + \varepsilon}} \right],$$

*where we use $\mathbf{v}_i^0 = 0$. Summing over $i = 1, 2, \cdots, d$ and $k = 1, 2, \cdots, K$, we have*

$$\sum_{k=1}^{K}\sum_{i=1}^{d} \mathbb{E}_{\mathcal{F}_{k-1}} \left[ \sqrt{\widetilde{\mathbf{v}}_i^k + \varepsilon} \right] \leq K\|\boldsymbol{\sigma}\|_1 + Kd\sqrt{\varepsilon} + \sum_{k=1}^{K}\sum_{t=1}^{k}\sqrt{\beta^{k-t}}(1-\beta)\sum_{i=1}^{d}\mathbb{E}_{\mathcal{F}_{t-1}} \left[ \frac{|\nabla_i f(\mathbf{x}^t)|^2}{\sqrt{\widetilde{\mathbf{v}}_i^t + \varepsilon}} \right]$$

$$= K\|\boldsymbol{\sigma}\|_1 + Kd\sqrt{\varepsilon} + \sum_{t=1}^{K}\sum_{k=t}^{K}\sqrt{\beta^{k-t}}(1-\beta)\sum_{i=1}^{d}\mathbb{E}_{\mathcal{F}_{t-1}} \left[ \frac{|\nabla_i f(\mathbf{x}^t)|^2}{\sqrt{\widetilde{\mathbf{v}}_i^t + \varepsilon}} \right]$$

$$\leq K\|\boldsymbol{\sigma}\|_1 + Kd\sqrt{\varepsilon} + \frac{1-\beta}{1-\sqrt{\beta}}\sum_{t=1}^{K}\sum_{i=1}^{d}\mathbb{E}_{\mathcal{F}_{t-1}} \left[ \frac{|\nabla_i f(\mathbf{x}^t)|^2}{\sqrt{\widetilde{\mathbf{v}}_i^t + \varepsilon}} \right]$$

$$= K\|\boldsymbol{\sigma}\|_1 + Kd\sqrt{\varepsilon} + (1+\sqrt{\beta})\sum_{t=1}^{K}\sum_{i=1}^{d}\mathbb{E}_{\mathcal{F}_{t-1}} \left[ \frac{|\nabla_i f(\mathbf{x}^t)|^2}{\sqrt{\widetilde{\mathbf{v}}_i^t + \varepsilon}} \right].$$

**Lemma 6** *When each entry of $\mathbf{x} \in \mathbb{R}^d$ is generated from Gaussian distribution with zero mean and unit variance, we have $\mathbb{E}\left[\|\mathbf{x}\|_1\right] \geq \sqrt{\frac{2d}{\pi}}\mathbb{E}\left[\|\mathbf{x}\|_2\right]$.*

**Proof 7** *When $\mathbf{x}_i \sim \mathcal{N}(0,1)$, we have*

$$\mathbb{E}\left[|\mathbf{x}_i|\right] = \sqrt{\frac{2}{\pi}}, \quad \mathbb{E}\left[\mathbf{x}_i^2\right] = 1,$$

$$\mathbb{E}\left[\|\mathbf{x}\|_1\right] = \sum_{i=1}^{d}\mathbb{E}\left[|\mathbf{x}_i|\right] = d\sqrt{\frac{2}{\pi}},$$

$$\mathbb{E}\left[\|\mathbf{x}\|_2^2\right] = \sum_{i=1}^{d}\mathbb{E}\left[\mathbf{x}_i^2\right] = d,$$

$$\mathbb{E}\left[\|\mathbf{x}\|_2\right] = \mathbb{E}\left[\sqrt{\|\mathbf{x}\|_2^2}\right] \overset{(1)}{\leq} \sqrt{\mathbb{E}\left[\|\mathbf{x}\|_2^2\right]} = \sqrt{d},$$

$$\frac{\mathbb{E}\left[\|\mathbf{x}\|_1\right]}{\mathbb{E}\left[\|\mathbf{x}\|_2\right]} \geq \sqrt{\frac{2d}{\pi}}.$$

*where we use the concavity of $\sqrt{x}$ in $\overset{(1)}{\leq}$.*

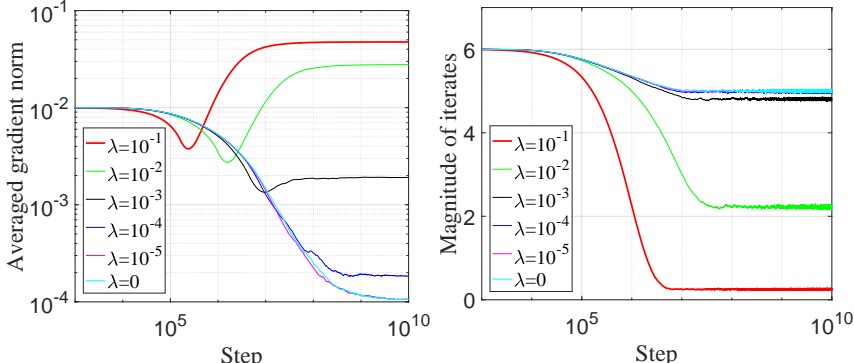

Figure 5: Illustrations of $\frac{1}{k}\sum_{t=1}^{k}|\nabla f(x^t)|$ (left) and $x^k$ (right) over steps on the toy example.

## D A Toy Example with Large $\lambda$

Consider the following function:

$$f(x) = \frac{(x-x^*)^2}{200}, \text{ with the stochastic gradient } g(x) = \begin{cases} x - x^* - 1, & \text{with probability } p = 0.1, \\ -\frac{1}{10}(x - x^* - \frac{10}{9}), & \text{with probability } 1 - p. \end{cases}$$

We set $K = 10^{10}$, $\theta = 1 - \frac{1}{\sqrt{K}}$, $\beta = \sqrt{\theta}$, $\eta = \frac{1}{\sqrt{K}}$, $\varepsilon = 10^{-10}$, $m^0 = 0$, $v^0 = 0$, and $x^1 = x^* + 1$ for AdamW, where $x^* = 5$ is the minimum solution of $f(x)$. We test $\lambda = \{10^{-1}, 10^{-2}, 10^{-3}, 10^{-4}, 10^{-5}, 0\}$ such that $x^* < \frac{1}{\lambda}$ and thus the KKT conditions (2) reduce to $|\nabla f(x^*)| = 0$ at the minimum solution. So we can use the gradient norm $|\nabla f(x)|$ to measure the convergence. From Figure 5, we see that AdamW fails to converge to $x^*$ when $\lambda = \{10^{-1}, 10^{-2}, 10^{-3}\}$, indicating that large values of $\lambda$ exceeding a certain threshold may cause AdamW neither to converge to the minimum solution nor to a KKT point satisfying (2)[3]. In practical implementations, excessively large values of $\lambda$ are typically avoided, as they may drive the parameters toward zero and away from the minimum solution.

## E Experimental Details

In the main paper, we conduct several representative deep learning experiments to empirically support our claims, covering classic image classification and language processing tasks. For the vision tasks, we independently train ResNet50 [50] on CIFAR100 [51] and ImageNet [52] datasets; For the language task, we adopt the GPT-2 [53] architecture and pretrain it on the OpenWebText [54] dataset. Code is released at https://github.com/adonis-dym/Convergence-Rate-AdamW .

Our experiments involve the computation of the full training loss $f(\mathbf{x}^k)$ as well as the full gradient $\nabla f(\mathbf{x}^k)$. However, in the typical stochastic training paradigm, one often updates the parameter $\mathbf{x}^k$ on-the-fly immediately after obtaining the stochastic gradient $\mathbf{g}^k$ from the backward pass. To get an accurate measurement and avoid interfering with the normal training process, we propose to split each epoch into two separate phases: *training phase* and *logging phase*. In the training phase, we traverse the dataset once with stochastic updates, where the model parameters are updated upon processing each mini-batch. In the logging phase, we conduct a second traversal over the training dataset while keeping the model parameters frozen. Since the loss function is typically defined to be the average over all training samples and the gradient computation is inherently linear, we accumulate the losses and stochastic gradients across mini-batches during this phase. This yields the exact values of the full training loss $f(\mathbf{x}^k)$ and full gradient $\nabla f(\mathbf{x}^k)$ at the current iteration.

In the following, we detail each experimental setup individually:

*i) ResNet50 - CIFAR100:* CIFAR100 is a simple benchmark dataset that is widely used for quick and efficient evaluation of deep learning tasks. It contains a training split of 50000 examples and a test split of 10000 examples, although we do not perform evaluation on the test set in this work. Following the official implementation, we use the `torch.optim.AdamW` API to configure the optimizer. We initialize the learning rate to $3 \times 10^{-3}$, train the ResNet50 model for 100 epochs, and apply a cosine learning rate decay schedule during the whole training process. Setting the batch size to 128, each epoch consists of $\lfloor 50000/128 \rfloor + 1 = 391$ steps, where the additional step accounts for the final truncated batch which contains the remaining samples. The total number of

---

[3]This does not conflict with [6] because [6] only considered deterministic AdamW.

steps is $K = 391 \times 100 = 39100$. Without loss of generality, we compute the noise vector $\boldsymbol{\sigma}^k = \mathbf{g}^k - \nabla f(\mathbf{x}^k)$ using the stochastic gradient $\mathbf{g}^k$ obtained from the first batch at the logging phase. We leave the weight decay $\lambda$ as its default value 0.01, and complete the training task with a single NVIDIA A100 GPU.

*ii) ResNet50 - ImageNet:* To evaluate the scalability of our conclusions on larger-scale dataset, we conduct experiments on the ImageNet dataset using the same ResNet50 architecture. ImageNet consists of approximately 1.28 million training images and 50,000 validation images across 1,000 classes, which also come with an official dataset split. We employ the training script from PyTorch Image Models (`timm`) [55], making only the necessary modifications to suit our experimental setup. We adopt the same optimizer configuration as previously, but compute the noise vector using the last batch at the logging phase, as the `timm` script discards incomplete batch and ensures uniform batch sizes. We follow the standard ImageNet training protocol for ResNet-50, which consists of 90 epochs as commonly adopted in the literature and official implementations [50, 55]. The first 10 epochs are used for learning rate linear warmup from 0 to $3 \times 10^{-3}$, followed by cosine decay over the remaining 80 epochs. We apply standard data augmentation techniques including RandAugment, Mixup (0.1), and CutMix (1.0). Setting the batchsize to 4096, each epoch consists of 312 minibatches and the total number of steps is $K = 28080$. We set $\lambda = 0.1$ and complete the training task using 8 NVIDIA A100 GPUs.

*iii) GPT2 - OpenWebText:* To assess the generality of our conclusions across different modalities, we further evaluate on a language modeling task using GPT-2. We pretrain this model on the OpenWebText dataset under the NVIDIA Megatron-LM codebase [56], which is a widely adopted framework for large-scale language model training. Unlike the previous settings, where computing the full training loss and gradient over the entire dataset is tractable, the OpenWebText dataset is substantially larger, containing approximately 9 billion tokens. Consequently, an entire pass through the dataset to get the full training loss $f(\mathbf{x}^k)$ and gradient $\nabla f(\mathbf{x}^k)$ is computationally infeasible. Instead, we approximate these quantities by accumulating their values over 100 consecutive mini-batches at the logging phase. We follow the Megatron-LM official GPT-2 training configuration with minimal modifications to suit our experimental needs. We train a GPT-2 Small model with approximately 125M parameters. The model is optimized using the fused implementation of AdamW from NVIDIA Apex package, which is the default setting in Megatron-LM. We set the learning rate to $3 \times 10^{-3}$ and weight decay to 0.05. Following the de facto standard in large-scale language model training, we use $(\theta, \beta) = (0.9, 0.95)$ (that is, the commonly used $(\beta_1, \beta_2)$) instead of the conventional $(0.9, 0.999)$ setting. The total training process runs for 50,000 iterations, where the learning rate is linearly warmed up for the first 2,000 iterations and then decayed following a cosine schedule. We set the global batch size to 640 and train the model for $K = 50000$ steps, and complete the training task using 8 NVIDIA A100 GPUs.

