# OpenReview forum: "On the $O(\frac{\sqrt{d}}{K^{1/4}})$ Convergence Rate of AdamW Measured by $\ell_1$ Norm"
_NeurIPS.cc/2025/Conference — NeurIPS 2025 poster_

### Official Review · Reviewer_bJjL · 2025-06-10

**Clarity:** 2
**Significance:** 2
**Originality:** 2
**Rating:** 4
**Confidence:** 3

**Summary:**

This paper investigates the convergence properties of AdamW under standard optimization settings. The convergence is analyzed with respect to the average of the L1-norm of the gradient across iterations. The assumptions made—smoothness of the objective function, unbiased stochastic gradients, and bounded gradient variance—are standard in the literature. Under these assumptions, the authors show that AdamW achieves convergence rates competitive with those of SGD.

The core contribution lies in the theoretical convergence analysis, along with a detailed comparison of assumptions and results relative to prior work. The authors claim theirs is the first convergence result for AdamW and Adam$^{1}$ established under these standard conditions.


$^{1}$ [1] provides similar convergence results for Adam under standard assumptions, although their analysis requires a conversion over optimality conditions, targets a different set of converged parameters, and involves compromises in the range of hyperparameters.

[1] Ahn, Kwangjun, and Ashok Cutkosky. "Adam with model exponential moving average is effective for nonconvex optimization." arXiv preprint arXiv:2405.18199 (2024).

**Questions:**

$\textbf{Technical quality:}$

1. Line 88: The stated rate $\mathcal{O}\left(\frac{\sqrt{d}}{\sqrt{K}}\right)$ is reminiscent of convergence guarantees for stochastic problems. However, is this rate optimal in the deterministic setting? Could the authors elaborate on why guarantees for stochastic settings can be directly extended to deterministic ones?


2. Line 136: You assume a relatively tight constraint on $\beta_2$ (i.e., $\beta_1 \leq\beta_2 \leq \frac{(1 + \beta_1)^{2}}{4}$), but one can derive similar bounds under the looser condition $\beta_1 \leq \beta_2 \leq \sqrt{\beta_1}$:
$$\frac{(1-\beta_{1})^{2}}{(1-\beta_{2})}\sum_{n=1}^{s}\left(\frac{\beta_{1}^2}{\beta_{2}}\right)^{s-1-n} = \frac{(1-\beta_{1})^{2}}{(1-\beta_{2})}\frac{1 - \left(\frac{\beta_{1}^2}{\beta_{2}}\right)^{s-1}}{1 - \frac{\beta_{1}^2}{\beta_{2}}} \leq \frac{(1-\beta_{1})^{2}}{(1-\beta_{2})^{2}}\leq \frac{(1- \sqrt{\beta_{1}})^{2}(1+ \sqrt{\beta_{1}})^{2}}{(1-\sqrt{\beta_{1}})^{2}}\leq 4.$$
For instance, supposing $\beta_{1} = 0.9$, then the range for $\beta_{2}$ changes from $(0.9, 0.9025)$ to $(0.9, 0.9486)$. This alternative formulation could potentially simplify the analysis and lead to similar bounds.

3. One of the core distinctions between Adam and AdamW is the additional weight decay term in AdamW. I assume that one of the main technical challenges lies in handling this term. However, it appears that simply controlling $\lambda$ allows the adaptation of Adam's existing analysis to AdamW with minimal changes, (intuitively). Could the authors clarify what specific technical hurdles they faced in extending the convergence analysis framework beyond tuning $\lambda$ to a particular range?

$\textbf{Other questions:}$

1. Regarding the AdamW update rule:, I believe the $\epsilon$ term is typically added outside the square root in the denominator. (But I assume it will not affect the analysis or negate the conclusion.). Additionally, I recommend adding the line: $g \leftarrow \text{GradOracle}(\cdot)$.

2. Figure 3: Please specify the model used in the experiments. I assume each subfigure corresponds to the same model trained with different values of $\lambda = 0.01, 0.05, 0.1$. However, it is unclear why the maximum norm is not monotonic with respect to $\lambda$—for example, $\lambda = 0.05$ yields the largest norm (~12.5). Can the authors provide insight into this behavior?

3. The paper repeatedly states that “∇f(x) is drawn from Gaussian distribution N (0, 1)”. While this is convenient for analysis, it may oversimplify the problem. In practice, the gradient distribution often exhibits drifting means, I believe.

**Ethical Concerns:**

["NO or VERY MINOR ethics concerns only"]

**Final Justification:**

I will keep my score due to the raised weaknesses 4 and 5.

**Limitations:**

yes

**Quality:**

2

**Strengths And Weaknesses:**

$\textbf{Strengths:}$

1. The paper provides a complete and rigorous convergence proof of AdamW under standard assumptions—namely, smoothness of the objective, unbiased gradient estimates, and bounded variance. As claimed by the authors, this is the first such result under these widely accepted conditions.


2. The exposition includes a reasonably clear summary of existing convergence results for both Adam and AdamW, helping position the current work within the broader literature.


$\textbf{Weakness:}$

1. The technical contributions in Section 3.1 are presented as key innovations; however, they are relatively standard. In particular, Lemma 2 follows known analysis patterns and could be further improved (as discussed in point 2 under the "Technical Quality" of  "Question"). The second main result is similarly derived by fixing $\lambda$ to a specific value, which limits its novelty.


2. The paper lacks sufficient interpretability of its technical derivations. For instance, some important steps in the proofs are not explained. Line 152 appears to rely on Jensen’s inequality, and line 3 of Proof 3 uses the Cauchy–Schwarz inequality, yet neither is explicitly stated. This omission makes it harder for readers to follow the reasoning.


3. The analysis omits the bias correction terms in AdamW. While this simplification may help theoretical tractability, such terms are critical for practical performance.


4. The theoretical results do not offer meaningful insight into Adam/AdamW’s empirical superiority, nor do they provide guidance for practical algorithm design. The convergence guarantees require hyperparameter settings—such as $\beta_1$, $\beta_2$, $\lambda$, and $\epsilon$—that deviate from those typically used in practice, which reduces the relevance of the results.


5. Overall, while the work fills a theoretical gap, its practical and conceptual significance remains unclear beyond establishing convergence under standard conditions (by setting extreme values for the hyperparameters, which potentially ``approximate'' SGD).

---

> ### Author Rebuttal · Authors · 2025-07-28
>
> We sincerely thank a lot for the reviewer's effort and valuable suggestions on our manuscript. Below are our responses to the major concern.
>
> $\mathbf{Response\enspace to\enspace Weaknesses}$:
>
> $\mathbf{Weakness\enspace 1}$.
>
> While the technical derivation in Section 3.1 is relatively straightforward, we highlight its importance in establishing the bound of term (g) (line 147) to constrain term (i) (line 154). While the techniques in Section 3.1 represent one aspect of our innovations, the key theoretical innovation of our proof appears in Section 3.2, where we strategically relax terms (f) and (g) (line 151) and control the second moment estimate via Eq. (12) (line 153). Then we effectively absorb term (h) into the relaxed bound derived from term (f) (line 154). This sequence of operations enables us to manage the troublesome term (g) (line 129) introduced by weight decay and derive the bound on line 161. Finally, we eliminate the dependence on $\varepsilon$ in term (k) (line 163) by the tight bound of the second moment estimate in Eq. (12) and careful configuration of $\varepsilon$.
>
> $\mathbf{Weakness\enspace 2\enspace and\enspace 3}$.
>
> We are grateful to the reviewer for these constructive suggestions and will carefully incorporate detailed explanations of the technical derivations in our final version. Specifically, we will try to add a new theorem establishing convergence guarantees for the complete AdamW with bias correction terms in the Appendix.
>
> $\mathbf{Weakness\enspace 4\enspace and\enspace 5}$.
>
> (1). Insight into Adam/AdamW’s empirical superiority.
>
> The reviewer has raised several valuable research questions, for example, offering insight into Adam/AdamW’s empirical superiority, which indeed is a fundamentally important challenge worthy of dedicated investigation in a separate paper (e.g., [1-6]), and it falls beyond the scope of our current study. Our paper establishes the convergence rate of AdamW, which is analogous to that of SGD, through the lens in the traditional optimization field.
>
> (2). Provide guidance for practical algorithm design.
>
> Designing practical algorithms for LLMs training is also a fundamentally important challenge worthy of dedicated investigation in a separate paper. While numerous optimizers are proposed annually, some marginally outperforming AdamW while others failing to deliver claimed improvements, AdamW remains the dominant optimizer for LMMs to date. Providing novel and effective algorithmic design principles presents significant challenges and it is also beyond the scope of this paper.
>
> (3). Hyperparameter settings that deviate from those typically used in practice.
>
> There is a gap between theory and practice in hyperparameter configuration. Indeed, modern LLMs are not trained to full convergence, and practical configurations often deviate from theoretical settings. For instance, prior work [7,8] demonstrates through constructed examples that Adam with common hyperparameters ($\beta_1=0.9$, $\beta_2={0.999, 0.997, 0.995, 0.993}$, and step-size $\eta_k=0.1/\sqrt{k}$) fail to converge to stationary points (see [8, Figure 2]). Similarly, while nonconvex SGD analysis typically assumes step-sizes decaying to zero ($\eta_k=1/\sqrt{k}$), practical LLMs training often employs cosine decay scheduler that plateau at small positive values. We fully agree that investigating Adam's property under realistic configurations represents an important research direction, as it could yield valuable insights for Adam/AdamW’s empirical superiority and practical algorithm design. However, the practical configurations may not guarantee the convergence. For example, empirical evidence from recent study [9, Figure 9] demonstrates that during the successful training of BERT and GPT-2, the $\ell_2$ gradient norm typically remains within the range [0,10] and hardly decreases throughout the optimization process.
>
> (4). Extreme values for the hyperparameters potentially approximating SGD.
>
> While our analysis employs standard convergence proof methodologies in the traditional optimization field, it exhibits fundamental difference from that of SGD, although certain hyperparameter choices were made primarily for theoretical tractability. See the key theoretical innovations in the response to Weakness 1. We sincerely thank the reviewer for recognizing that our work fills a theoretical gap and we maintain that this theoretical contribution meets NeurIPS' rigorous theoretical standards.
>
> [1]. Y. Zhang, et al. Why transformers need Adam: A Hessian perspective. NeurIPS 2024.
>
> [2]. F. Kunstner, et al. Noise is not the main factor behind the gap between SGD and Adam on transformers, but sign descent might be. ICLR 2023.
>
> [3]. F. Kunstner, et al. Heavy-tailed class imbalance and why Adam outperforms gradient descent on language models. NeurIPS 2024.
>
> [4]. T. Sreckovic, et al. Is your batch size the problem? Revisiting the Adam-SGD gap in language modeling. arXiv 2506.12543, 2025.
>
> [5]. A. Tomihari, et al. Understanding why Adam outperforms SGD: gradient heterogeneity in transformers. arXiv 2502.00213, 2025.
>
> [6]. Y. Pan, et al. Toward understanding why Adam converges faster than SGD for transformers. arXiv 2306.00204, 2023.
>
> [7]. Y. Zhang, et al. Adam can converge without any modification on update rules. NeurIPS 2024.
>
> [8]. B. Wang, et al. Provable adaptivity of Adam under non-uniform smoothness. KDD 2024.
>
> [9]. H. Tran, et al. Empirical tests of optimization assumptions in deep learning. arXiv 2407.01825, 2024
>
> $\mathbf{Response\enspace to\enspace Questions}$:
>
> $\mathbf{Technical\enspace Quality\enspace 1}$.
>
> Recent work [10, Theorem 4.1] established a fundamental lower bound $ \Omega\left( \sqrt{ \frac{dL ( f(x^1)-f^* )}{T} }+ \left( \frac{ dL ( f(x^1)-f^* ) ||\sigma||_1^2 }{T} \right)^{1/4} \right) $ for SGD with $\ell_1$-norm gradient measurement, which implies that the $O( \sqrt{ \frac{d}{K} } )$ rate is optimal for GD under identical $\ell_1$-norm criteria in the determnistic setting.
>
> When the noise variance is small satisfying $\sigma_s^2 \leq \frac{ L( f(x^1)-f^* ) }{ K }$, the first term in our convergence rate (line 65) becomes identical to the second term, thus making the convergence rate for stochastic settings directly extended to deterministic ones (or, more broadly speaking, low-noise regimes).
>
> [10]. R. Jiang, et al. Convergence analysis of adaptive gradient methods under refined smoothness and noise assumptions. COLT 2025.
>
> $\mathbf{Technical\enspace Quality\enspace 2}$.
>
> We gratefully thank the reviewer for this looser condition. We will add an acknowledgment footnote in our final version to clarify that this condition is derived by the NeurIPS reviewer.
>
> $\mathbf{Technical\enspace Quality\enspace 3}$.
>
> As explained in the response to Weakness 1, our key theoretical innovations are as follows:
>
> First, we strategically relax terms (f) and (g) (line 151) and control the second moment estimate via Eq. (12) (line 153), which enables the novel absorption of term (h) into the relaxed bound derived from term (f) (line 154). This sequence of operations enables us to manage the troublesome term (g) (line 129) introduced by the weight decay and derive the bound on line 161.
>
> Second, we eliminate the dependence on $\varepsilon$ in term (k) (line 163) by careful configuration of $\varepsilon$ and tight bound of the second moment estimate in Eq. (12).
>
> The first step is crucial to deal with the troublesome weight decay term in AdamW and this idea has no precedent in the literature. We respectfully disagree with the reviewer that they are minimal and intuitively changes.
>
> For Adam, while the first step becomes unnecessary (because weight decay disappears in Adam), the second step remains essential for establishing tight $\varepsilon$-independent convergence rate. Without our two key techniques in the second step, the closest related work [11] achieves only slower and $\varepsilon$-dependent convergence rate. See the detailed comparisons to [11] on lines 204-211. This is the principal technical hurdle existing literature on Adam faced to achieve tight convergence rate comparable to SGD.
>
> [11]. H. Li, et al. Convergence of Adam under relaxed assumptions. NeurIPS 2023.
>
> $\mathbf{Other\enspace Question\enspace 1}$.
>
> The reviewer is absolutely correct to note that in the standard PyTorch implementation, $\varepsilon$ is added outside the square root (see Lines 208-209 in our paper). We placed this term inside the square root to simplify the mathematical expressions and it does not affect our conclusion. This is also adopted in some theoretical literature [12,13].
>
> $g\leftarrow \mbox{GradOracle}(\cdot)$: we gratefully thank the reviewer for this suggestion and we will add this line in the algorithm representation of our final version.
>
> [12]. A. Defossez, et al. A simple convergence proof of Adam and AdaGrad. TMLR 2022.
>
> [13]. S. Xie, et al. Adam exploits $\ell_{\infty}$-geometry of loss landscape via coordinate-wise adaptivity. ICLR 2025.
>
> $\mathbf{Other\enspace Question\enspace 2}$.
>
> We have specified the models on the top of Figure 1. In Figures 1-4, the first column presents the experimental results of ResNet50 on CIFAR100, the second column is ResNet50 on ImageNet, and the third column is GPT2 on OpenWebText. For each model-dataset combination, we adopted the standard $\lambda$ values commonly used in the literature. Because each subfigure represents distinct model-dataset pairs, the maximum norm does not exhibit monotonic behavior with respect to $\lambda$.
>
> $\mathbf{Other\enspace Question\enspace 3}$.
>
> We fully agree with the reviewer. The gradient distribution often exhibits drifting means across different layers/blocks in modern architectures like Transformers. We conducted experiments on real-world deep learning tasks (Figure 2), which reveal that the ratio $\frac{ || \nabla f(x) ||_1 }{ || \nabla f(x) ||_2 } \approx \frac{ \sqrt{d} }{10} $, smaller than the theoretical prediction of $\sqrt{d}$.

---

> ### Comment · Reviewer_bJjL · 2025-08-03
>
> Thank you for your response. I have read through the rebuttal and have decided to keep my rating.

---

> > ### Author Response · Authors · 2025-08-06
> >
> > We sincerely thank you to review our rebuttal. Thank you again most kindly for the looser condition.

---

### Official Review · Reviewer_9v39 · 2025-07-01

**Clarity:** 3
**Significance:** 3
**Originality:** 3
**Rating:** 6
**Confidence:** 4

**Summary:**

The paper proves that AdamW with a constant weight-decay parameter λ achieves a non-convex convergence rate $\frac1K\sum_{k=1}^K \mathbb E\bigl[\|\nabla f(x_k)\|_1\bigr] = O \Bigl(\tfrac{\sqrt d}{K^{1/4}}\Bigr)$,
matching the best known $K^{-1/4}$ rate of SGD up to a $\sqrt d$ factor. The analysis avoids the vanishing-$\lambda$ assumption common in prior work on Adam-type methods and introduces a neat decoupling of the EMA constants $\beta, \theta$ from weight decay. A corollary shows that standard Adam ($λ = 0$) meets the same rate without the restrictive $\beta–\theta$ coupling used in earlier proofs. The authors also provide empirical evidence that $\|\nabla f\|_1\approx\Theta(\sqrt d)\|\nabla f\|_2$, clarifying why an $\ell_1$ metric is theoretically comparable to SGD’s $\ell_2$ guarantee.

**Questions:**

- Although I am positive about the paper overall, I am still not sure why $\ell_1$-norm bound is adopted for the convergence analysis. Is an analogous $\ell_2$ or averaged-iterate result attainable with the same technique, or is $\ell_1$ essential?

- Could the author include a study varying $\lambda$ to show where theory-suggested upper bounds start to matter?

**Ethical Concerns:**

["NO or VERY MINOR ethics concerns only"]

**Final Justification:**

Since the authors have addressed most of my concerns during the rebuttal period, and I want to support this paper for the acceptance, I have increased my score from 5 to 6.

**Limitations:**

- The paper itself acknowledges that the step-size/$\lambda$ regime is mainly of theoretical interest; practical recipes (e.g., linear warm-up or cosine decay) are not analyzed. Suggest adding a short discussion of how to bridge this gap or empirical traces showing safe hyper-parameter inflation.

- There are some typos (e.g., $1 \leq \beta \leq 1$ in Corollary 2), but they are easily fixable.

**Paper Formatting Concerns:**

There are no formatting issues.

**Quality:**

3

**Strengths And Weaknesses:**

## Strengths

1. First constant-$\lambda$ analysis for AdamW: Earlier theory required $\lambda \to 0$ or elaborate scheduling; this work closes that gap and aligns theory with common practice.

2. Tight rate in an $\ell_1$ norm: By linking $\ell_1$ and $\ell_2$ gradient norms in high-dimensional settings, the paper shows its bound is essentially on par with the optimal SGD rate.

3. Clean proof techniques: The authors streamline variance-control lemmas and sidestep bounding tricks (e.g., biased-second-moment correction) that earlier analyses relied on, making the argument easier to follow or reuse.

## Weaknesses

1. Practical hyper-parameter ranges are narrow. The theorem fixes $\(\eta=\tfrac{\sqrt{f(x_1)-f^\*}}{4dKL}\)$, which can be orders of magnitude below learning-rates used in LLM training, and imposes conservative upper bounds on $\lambda$. This limits immediate applicability.

2. There are some minor presentation flaws: A few typographical lapses remain (e.g., $\beta$ range in Corollary 2 printed as $1 \leq \beta \leq 1$) and some constants are undeclared in the main text; these are easily fixed in revision.

## Other comments

- The authors should include the recent paper below [1], which is related to the convergence analysis of Adam-style optimizer.

[1] ADOPT: Modified Adam Can Converge with Any β2 with the Optimal Rate. https://arxiv.org/abs/2411.02853

---

> ### Author Rebuttal · Authors · 2025-07-28
>
> We sincerely thank a lot for the reviewer's effort and valuable suggestions on our manuscript. Below are our responses to the major concern.
>
> $\mathbf{Response\enspace to\enspace Weaknesses}$:
>
> $\mathbf{Weakness\enspace 1}$.
>
> We acknowledge that some hyper-parameters are unpractical, such as $\eta=\sqrt{ \frac{ f(x^1)-f^* }{ 4KdL }}$. We would like to emphasize the inherent challenges in bridging theoretical analysis with practical implementations. Indeed, modern LLMs are not trained to full convergence, and practical configurations often deviate from theoretical settings. For instance, prior work [1,2] demonstrates through constructed examples that Adam with common hyperparameters ($\beta_1=0.9$, $\beta_2={0.999, 0.997, 0.995, 0.993\}$, and step-size $\eta_k=0.1/\sqrt{k}$) fail to converge to stationary points (see [2, Figure 2]). Similarly, while nonconvex SGD analysis typically assumes small step-sizes decaying to zero ($\eta_k=O(\frac{1}{\sqrt{k}})$), practical LLMs training often employs cosine decay scheduler that plateau at small positive values. We fully agree that investigating Adam's property under realistic configurations represents an important research direction, as it could yield valuable insights for LLMs hyperparameter tuning and training dynamics interpretation. However, the practical configurations may not guarantee the convergence. For example, empirical evidence from recent study [3, Figure 9] demonstrates that during the successful training of BERT and GPT-2, the $\ell_2$ gradient norm typically remains within the range [0,10] and hardly decreases throughout the optimization process, even though the objective function decreases sufficiently.
>
> We will follow the reviewer's valuable suggestion to conduct dedicated research to bridge this theory-practice gap in future work.
>
> [1]. Y. Zhang, et al. Adam can converge without any modification on update rules. NeurIPS 2024.
>
> [2]. B. Wang, et al. Provable adaptivity of Adam under non-uniform smoothness. KDD 2024.
>
> [3]. H. Tran, et al. Empirical tests of optimization assumptions in deep learning. arXiv 2407.01825, 2024
>
> $\mathbf{Weakness\enspace 2}$.
>
> We are grateful to the reviewer for these presentation flaws and will carefully address them in our final version.
>
> $\mathbf{Other\enspace comments}$.
>
> We also sincerely thank the reviewer for bringing this relevant citation to our attention. It will be properly cited in our revised manuscript.
>
> $\mathbf{Response\enspace to\enspace Questions}$:
>
> $\mathbf{Question\enspace 1}$.
>
> We appreciate this insightful question regarding our use of the $\ell_1$ norm. Our justification is threefold:
>
> First, since Adam can be viewed as a smoothed variant of SignSGD/Signum [4,5], both of which employ $\ell_1$ norm gradient measurements. Thus, adopting the $\ell_1$ norm for Adam's convergence analysis represents a consistent choice.
>
> Second, as discussed on lines 19-23, recent work [6] established that Adam's iterates converge to solutions of problem (1) under the infinity norm constraint. The resulting KKT conditions (Eq. 2) naturally involve the gradient's $\ell_1$ norm. Then, we explained why we subsequently simplify the KKT conditions to the pure gradient $\ell_1$ norm on lines 28-32.
>
> Third, when using the gradient $\ell_2$ norm, it would yield slower convergence rate as demonstrated in [7] (see the comparisons on lines 204-211). While our techniques could theoretically improve [7]'s rate, this would require the impractical condition $\varepsilon\geq \sigma_s^2$. In contrast, our $\ell_1$-norm approach only needs $\varepsilon= \frac{ \sigma_s^2 }{d}$, a significantly milder requirement that aligns with common practice.
>
> [4]. J. Bernstein, et al. SignSGD: compressed optimisation for non-convex problems. ICML 2018.
>
> [5]. L. Balles, et al. Dissecting Adam: The sign, magnitude and variance of stochastic gradients. ICML 2018.
>
> [6]. S. Xie, et al. Implicit bias of AdamW: $\ell_{\infty}$ norm constrained optimization. ICML 2024.
>
> [7]. H. Li, et al. Convergence of Adam under relaxed assumptions. NeurIPS 2023.
>
> $\mathbf{Question\enspace 2}$.
>
> Following the reviewer's valuable suggestion, we have conducted additional experiments to evaluate the training behavior across different $\lambda$ values. We perform the analysis on both ResNet-50 (CIFAR100) and GPT-2 (OpenWebText), study the training behavior, and estimate the value of the derived upper bound of $\lambda$ to assess its compatibility with practical choices. For each experiment, we vary $\lambda$ values as ${0.01, 0.05, 0.1, 0.5}$ and report the average training loss at the beginning, end, and several intermediate epochs/steps. To provide an accurate yet computationally feasible estimation of the upper bound, we set $\hat{\sigma}_s^2 = \sigma_s^2$ and approximate the noise variance as
>
> $\sigma_s^2\approx\max_{k=1}^K ||g^k - \nabla f(x^k)||^2$.
>
> Similarly, the Lipschitz smooth constant is approximated as
>
> $L\approx\max_{k=1}^K \frac{||\nabla f(x^{k+1}) - \nabla f(x^k)||}{||x^{k+1} - x^k||}$.
>
> Additionally, we set $f^\star = 0$ to further ensure a conservative estimate. To obtain the best estimation of $\sigma_s^2$ and $L$, we compute the maximum over all $\lambda$ settings.  Given the knowledge of $d$, $K$, and $f(x^1)$, the required $\lambda$ bound can be computed. Specifically, for ResNet-50 (CIFAR100), we have the estimated upper bound of $0.5729$. For GPT-2 (OpenWebText), we estimate as $0.3464$. Below are the detailed numerical results.
>
> - ResNet-50 (CIFAR100)
> | Epoch         | 1 | 20 | 40 | 60 | 80 | 100 |
> |--------------|----|-----|-----|-----|-----|------|
> | $\lambda$=0.01 | 3.92237 | 0.55829 | 0.09655 | 0.02322 | 0.00325 | 0.00103 |
> | $\lambda$=0.05 | 3.95005 | 0.63125 | 0.22489 | 0.09712 | 0.00893 | 0.00120 |
> | $\lambda$=0.1  | 3.90191 | 0.93962 | 0.53329 | 0.21229 | 0.02505 | 0.00218 |
> | $\lambda$=0.5  | 3.87442 | 1.98060 | 1.78694 | 1.46280 | 0.76590 | 0.17032 |
>
> - GPT-2 (OpenWebText)
> | Step         | 1 | 10000 | 20000 | 30000 | 40000 | 50000 |
> |--------------|----|--------|--------|--------|--------|--------|
> | $\lambda$=0.01 | 7.05598 | 3.08517 | 3.02195 | 2.99443 | 2.98804 | 2.99229 |
> | $\lambda$=0.05 | 7.05515 | 3.09249 | 3.02159 | 2.97827 | 2.95811 | 2.96719 |
> | $\lambda$=0.1  | 7.06873 | 3.11096 | 3.03222 | 2.97760 | 2.94311 | 2.95332 |
> | $\lambda$=0.5  | 7.05931 | 3.27434 | 3.16656 | 3.06161 | 2.96586 | 2.96249 |
>
> These results indicate that AdamW exhibits stable convergence behavior even under relatively large $\lambda$ values, though it converges more slowly when $\lambda$ approaches the upper bound for ResNet.
>
> $\mathbf{Response\enspace to\enspace Limitations}$:
>
> $\mathbf{Limitation\enspace 1}$.
>
> We sincerely appreciate this insightful suggestion regarding the theory-practice gap in step-size selection. We fully agree that bridging our theoretical analysis with practical step-size regimes represents valuable future work. We will add a short discussion in Section 2.5 of the final version.
>
> $\mathbf{Limitation\enspace 2}$.
>
> We are grateful to the reviewer for catching the typos and we will correct it in our final version.

---

> > ### Comment · Reviewer_9v39 · 2025-08-05
> >
> > Thank you for your response.
> >
> > It addressed most of my concerns, so I have increased my score from 5 to 6.

---

> > > ### Author Response · Authors · 2025-08-05
> > >
> > > We sincerely appreciate the time you have taken to review our rebuttal and are truly grateful for your favorable feedback. Thank you once again for your generous support.

---

### Official Review · Reviewer_pXdm · 2025-07-02

**Clarity:** 3
**Significance:** 3
**Originality:** 3
**Rating:** 4
**Confidence:** 4

**Summary:**

This paper provides a comprehensive convergence analysis for AdamW under non-convex smooth optimization. The convergence rate is optimal with respect to the iteration number, noise level, the initial function value gap, and the smooth parameter.

**Questions:**

I do not have any further questions.

**Ethical Concerns:**

["NO or VERY MINOR ethics concerns only"]

**Final Justification:**

Some of my concerns have been addressed by the additional experiments. However, the somewhat unsatisfactory setup of $\lambda$ and the polynomial dependency of $\epsilon$ have not been well addressed. I maintain a positive review.

**Limitations:**

yes

**Quality:**

3

**Strengths And Weaknesses:**

**Strengths:**

The major contribution of this paper is mainly in the theoretical part.

a. The convergence result for AdamW in this paper looks novel and sound. It seems that this is the first theoretical convergence result for AdamW with an explicit form of convergence rate. I check the parameter setups in detail, and most of the setups are reasonable. Although there are some gaps between the theoretical setups and the practical ones, it's acceptable for a theoretical paper.

b. The proof sketch is clearly written. There are some novel techniques to handle the additional decay parameter brought by AdamW.

**Weaknesses:**

a. For the detailed parameter setups, I have some concerns. First, as the authors also claim in Section 4.1, the setup for $\lambda$ can exceed the default $0.01$. Specifically, if $d$ is sufficiently large, e.g., the dimension of GPT-3.5, the requirement for $\lambda$ is not very satisfactory. I also find that $\lambda$ for experiments in this paper still applies $0.01$ or $0.1$. It's better to provide some experiments to verify that a large $\lambda$ is also acceptable.

b. The dependency of $\epsilon$ in the convergence rate seems to be $O(1/\epsilon)$. The authors then use a sufficiently small $1-\theta$ and $\eta$, namely $O(\epsilon)$ order to eliminate the factor $O(1/\epsilon)$. This may not be so satisfactory. There are some convergence results for Adam with a $O(\log(1/\epsilon))$ dependency. Is it possible to achieve the same dependency for AdamW? Also, the setup for $\epsilon$ is determined by $d$ instead of any constant like the setup in previous literature.

c. I think that a key difference of AdamW to Adam is to hand to the term $\lambda \\|x_i\\|_{\infty}$. The key solution is Lemma 3. So I advise the author to provide the detailed version of Lemma 3 in the main body of the paper.

---

> ### Author Rebuttal · Authors · 2025-07-28
>
> We sincerely thank a lot for the reviewer's effort and valuable suggestions on our manuscript. Below are our responses to the major concern.
>
> $\mathbf{Response\enspace to\enspace Weaknesses}$:
>
> $\mathbf{Weakness\enspace 1}$.
>
> In Theorem 1, our convergence rate holds for all $\lambda$ satisfying $\lambda\leq \frac{ \sqrt{d} }{ \sqrt{72} K^{3/4} } \sqrt[4]{ \frac{L^3}{\hat\sigma_s^2 (f(x^1)-f^*)} }$. As the dimension $d$ increases, it is reasonable to expect that the upper bound on the right-hand side grows accordingly, making it more likely to exceed 0.01 or 0.1, showing that our theory covers commonly used practical configurations.
>
> Following the reviewer's valuable suggestion, we have conducted additional experiments to evaluate the training behavior across different $\lambda$ values, especially large values. We perform the analysis on both ResNet-50 (CIFAR100) and GPT-2 (OpenWebText), study the training behavior, and estimate the value of the derived upper bound of $\lambda$ to assess its compatibility with practical choices. For each experiment, we vary $\lambda$ values as ${0.01, 0.05, 0.1, 0.5}$ and report the average training loss at the beginning, end, and several intermediate epochs/steps. To provide an accurate yet computationally feasible estimation of the upper bound, we set $\hat{\sigma}_s^2 = \sigma_s^2$ and approximate the noise variance as
>
> $\sigma_s^2\approx\max_{k=1}^K ||g^k - \nabla f(x^k)||^2$.
>
> Similarly, the Lipschitz smooth constant is approximated as
>
> $L\approx\max_{k=1}^K \frac{||\nabla f(x^{k+1}) - \nabla f(x^k)||}{||x^{k+1} - x^k||}$.
>
> Additionally, we set $f^\star = 0$ to further ensure a conservative estimate. To obtain the best estimation of $\sigma_s^2$ and $L$, we compute the maximum over all $\lambda$ settings.  Given the knowledge of $d$, $K$, and $f(x^1)$, the required $\lambda$ bound can be computed. Specifically, for ResNet-50 (CIFAR100), we have the estimated upper bound of $0.5729$. For GPT-2 (OpenWebText), we estimate as $0.3464$. Below are the detailed numerical results.
>
> - ResNet-50 (CIFAR100)
> | Epoch         | 1 | 20 | 40 | 60 | 80 | 100 |
> |--------------|----|-----|-----|-----|-----|------|
> | $\lambda$=0.01 | 3.92237 | 0.55829 | 0.09655 | 0.02322 | 0.00325 | 0.00103 |
> | $\lambda$=0.05 | 3.95005 | 0.63125 | 0.22489 | 0.09712 | 0.00893 | 0.00120 |
> | $\lambda$=0.1  | 3.90191 | 0.93962 | 0.53329 | 0.21229 | 0.02505 | 0.00218 |
> | $\lambda$=0.5  | 3.87442 | 1.98060 | 1.78694 | 1.46280 | 0.76590 | 0.17032 |
>
> - GPT-2 (OpenWebText)
> | Step         | 1 | 10000 | 20000 | 30000 | 40000 | 50000 |
> |--------------|----|--------|--------|--------|--------|--------|
> | $\lambda$=0.01 | 7.05598 | 3.08517 | 3.02195 | 2.99443 | 2.98804 | 2.99229 |
> | $\lambda$=0.05 | 7.05515 | 3.09249 | 3.02159 | 2.97827 | 2.95811 | 2.96719 |
> | $\lambda$=0.1  | 7.06873 | 3.11096 | 3.03222 | 2.97760 | 2.94311 | 2.95332 |
> | $\lambda$=0.5  | 7.05931 | 3.27434 | 3.16656 | 3.06161 | 2.96586 | 2.96249 |
>
> These results indicate that AdamW exhibits stable convergence behavior even under relatively large $\lambda$ values, though it converges more slowly when $\lambda$ approaches the upper bound for ResNet.
>
> $\mathbf{Weakness\enspace 2}$.
>
> (1) Use sufficiently small $1-\theta$ and $\eta$ to eliminate $ O(1/\varepsilon)$.
>
> From term (k) on line 163, our convergence rate depends on $\frac{1}{K} \sqrt[4]{ \frac{K}{\varepsilon} ( K ||\sigma||_1 + Kd \sqrt{\varepsilon} )^2 }$. When $d \sqrt{\varepsilon} \geq ||\sigma||_1$ (satisfied by $\varepsilon \geq \frac{\sigma_s^2}{d}$), the convergence rate becomes $\frac{1}{K} \sqrt[4]{ \frac{K}{\varepsilon} 4K^2d^2 \varepsilon }=O\left(\frac{\sqrt{d}}{K^{1/4}}\right)$ and we can eliminate the dependence on $\varepsilon$. Since practical implementations often use very small $\varepsilon$, we specifically choose $\varepsilon = \frac{\sigma_s^2}{d}$ to make $\varepsilon$ as small as possible while maintaining this condition. While larger values of $\varepsilon$ remain theoretically permissible, selecting values smaller than $\frac{\sigma_s^2}{d}$ would reintroduce explicit $\varepsilon$-dependence in the final convergence rate.
>
> So the crucial technique for eliminating the dependence on $\varepsilon$ in our convergence rate stems from the setting of $\varepsilon = \frac{\sigma_s^2}{d}$, rather than the specific choices of $1-\theta$ and $\eta$. From lines 352-355, we set $1-\theta$ to minimize the last two terms in Eq. (17), and determine $\eta$ and $\nu$ to balance the first two terms with the last two terms in Eq. (17), while strictly maintaining the constraint $\eta^2\leq \frac{\varepsilon(1-\theta)^2}{4L^2}$. When $\varepsilon$ increases, $\eta$, $\nu$, and $\lambda$ adjust proportionally.
>
> (2) Achieve the $O(\log(1/\varepsilon))$ dependency?
>
> To the best of our knowledge, existing convergence analyses for adaptive gradient methods can be broadly categorized into two frameworks. The first framework (e.g., [1-3] for Adam and [4] for RMSProp) achieves weak $\varepsilon$-dependence (typically $\log\frac{1}{\varepsilon}$) but often results in complex proofs with stronger dependence on terms $L$, $\sigma_s$, $f(x^1)-f^*$, and especially the dimension $d$ (only for Adam, excluding the simpler AdaGrad and RMSProp). The second framework (e.g., [5]) instead generally yields simpler proofs and tighter convergence rates but introduces explicit $\varepsilon$-dependence in the convergence rate (typically $1/\varepsilon^a$ for some $a$). Our work adopts the second framework. Currently, we do not know how to combine the advantages of the two frameworks to achieve the tight convergence rate while maintaining the $\log\frac{1}{\varepsilon}$ dependence on $\varepsilon$. Thanks for this insightful suggestion and we will investigate it in future work.
>
> (3) $\varepsilon$ is determined by $d$ instead of any constant.
>
> As discussed above, our analysis only requires $d \sqrt{\varepsilon} \geq ||\sigma||_1$ for $\varepsilon$, which is satisfied by $\varepsilon = \frac{\sigma_s^2}{d}$, making $\varepsilon$ only dependent on $\sigma_s$ and $d$, rather than the other constants.
>
> [1]. A. Defossez, et al. A simple convergence proof of Adam and AdaGrad. TMLR 2022.
>
> [2]. Y. Hong, et al. On convergence of Adam for stochastic optimization under relaxed assumptions. NeurIPS 2024.
>
> [3]. B. Wang, et al. Closing the gap between the upper bound and the lower bound of Adam’s iteration complexity. NeurIPS 2023.
>
> [4]. S. Xie, et al. Adam exploits $\ell_{\infty}$-geometry of loss landscape via coordinate-wise adaptivity. ICLR 2025.
>
> [5]. H. Li, et al. Convergence of Adam under relaxed assumptions. NeurIPS 2023.
>
> $\mathbf{Weakness\enspace 3}$.
>
> We are grateful to the reviewer for this constructive suggestion. We will move it in the main body of our final version.

---

> ### Comment · Reviewer_pXdm · 2025-08-07
>
> Thanks a lot for the author's reply, and sorry for the late reply. I appreciate the contribution of being the first convergence result for AdamW. I think that my major concern still remains largely due to the somewhat unsatisfactory setup of $\lambda$ and $\epsilon$.
>
> For $\lambda$,  it depends on a ratio between $\sqrt{d}$ and $K^{3/4}$. This ratio is highly dependent on the problem and the iteration number, which could be unstable for different training tasks.
>
> For $\epsilon$, since the authors use a similar framework as [Li et al., NeurIPS 2023], the dependency on $\epsilon$ has the same weakness as [Li et al., NeurIPS 2023], a polynomial dependency of $1/\epsilon$.
>
> I think that the rebuttal does not fully address my concern. However, as I claim in the preliminary review, this result is novel and interesting. I will keep my positive review.

---

> > ### Author Response · Authors · 2025-08-07
> >
> > We sincerely thank you to review our rebuttal and give the valuable feedback. We fully agree with the reviewer’s comments on the setup of $\lambda$ and $\varepsilon$. Regarding the parameter $\varepsilon$, both our results and [Li et al., NeurIPS 2023] have a polynomial dependency of $\frac{1}{\varepsilon}$. Our improvement is that we can eliminate the dependence on $\varepsilon$ by setting $\varepsilon=\frac{\sigma_s^2}{d}$, while [Li et al., NeurIPS 2023] requires $ \varepsilon=G^2\geq \sigma_s^2$ (see lines 209, 206, and 208 in our manuscript).
> >
> > We gratefully acknowledge the reviewer's constructive suggestions and will incorporate these insights to further enhance our results in future work. Thank you again most kindly.

---

### Official Review · Reviewer_FJAz · 2025-07-03

**Clarity:** 3
**Significance:** 2
**Originality:** 3
**Rating:** 4
**Confidence:** 3

**Summary:**

The authors proved a novel convergence rate of AdamW in terms of the minimal expected $L^1$-norm of the gradient, where the theoretical requirement for the hyperparameters almost fit the realities.

**Questions:**

1. Under a different set of conditions, [1] successfully derived a convergence rate of Adam, while $\beta_2  = 1-\tilde O(K^{-1})$ and is similar to yours. They claim that the proof can adapt to non-zero $\beta_1$  but hurts by having a factor of $\mathrm{poly}(1/(1-\beta_1))$. While in your corollary on the convergence rate of Adam, you have no requirement on $\beta_2$  and $\beta_1$  is close to $1$. Can you briefly compare how you improved this result?
2. Does it directly implies that the sequence converge to the KKT point? I am wondering if the infty norm of the trajectory is well controlled so that the gradient $L^1$ norm convergence implies the convergence to KKT points.

[1] Xie, S., Mohamadi, M. A., & Li, Z. (2024). Adam Exploits $\ell_\infty $-geometry of Loss Landscape via Coordinate-wise Adaptivity. arXiv preprint arXiv:2410.08198.

**Ethical Concerns:**

["NO or VERY MINOR ethics concerns only"]

**Final Justification:**

The authors have properly clarified the limitation and the strength of analysis by showing some related papers, and these limitaitons are    rather inherited from the whole line of the analysis. The main contribution in the analysis of the weight decay version of adam is still significant. Therefore, I raise my score based on careful consideration.

**Limitations:**

yes.

**Quality:**

3

**Strengths And Weaknesses:**

**Pros**

1. While being the most preferable optimizer in the era of large language models, AdamW is not well understood in terms of its theoretical properties like convergence and implicit biases. This work is the first one that proved the convergence rate of Adam with weight decay.
2. The content is well-organized and the result and its consequences are clearly stated. A

**Cons**

1. Some conditions on the hyperparameters are not realistic, indicating that the bound might not be sufficiently explaining  real-world optimization trajectories.
2. The theoretical results seem to provide little practical implications on tuning the hyperparameters and understandings on the dynamics, trejactories or implicit bias of AdamW. See questions.
3. The authors claim that the result is optimal in the sense that the corresponding 2-norm bound matchs the lower bound of SGD. However, the translation between 1-norm and 2-norm are semi-empirical.

---

> ### Author Rebuttal · Authors · 2025-07-28
>
> We sincerely thank a lot for the reviewer's effort and valuable suggestions on our manuscript. Below are our responses to the major concern.
>
> $\mathbf{Response\enspace to\enspace Weaknesses\enspace (Cons)}$:
>
> $\mathbf{Cons\enspace 1\enspace and\enspace 2}$.
>
> We agree with the reviewer's insightful comment regarding the theory-practice gap in hyperparameter configuration. Indeed, modern LLMs are not trained to full convergence, and practical configurations often deviate from theoretical settings. For instance, prior work [1,2] demonstrates through constructed examples that Adam with common hyperparameters ($\beta_1=0.9$, $\beta_2={0.999, 0.997, 0.995, 0.993\}$, and step-size $\eta_k=0.1/\sqrt{k}$) fail to converge to stationary points (see [2, Figure 2]). Similarly, while nonconvex SGD analysis typically assumes step-sizes decaying to zero ($\eta_k=O\left(1/\sqrt{k}\right)$), practical LLMs training often employs cosine decay scheduler that plateau at small positive values.
>
> We fully agree that investigating Adam's property under realistic configurations represents an important research direction, as it could yield valuable insights for LLMs hyperparameter tuning and training dynamics interpretation. However, the practical configurations may not guarantee the convergence. For example, empirical evidence from recent study [3, Figure 9] demonstrates that during the successful training of BERT and GPT-2, the $\ell_2$ gradient norm typically remains within the range [0,10] and hardly decreases throughout the optimization process, even though the objective function decreases sufficiently. So we maintain that our current work, which aligns with standard convergence proof methodologies in the traditional optimization field, satisfies NeurIPS' rigorous theoretical standards.
>
> [1]. Y. Zhang, et al. Adam can converge without any modification on update rules. NeurIPS 2024.
>
> [2]. B. Wang, et al. Provable adaptivity of Adam under non-uniform smoothness. KDD 2024.
>
> [3]. H. Tran, et al. Empirical tests of optimization assumptions in deep learning. arXiv 2407.01825, 2024.
>
> $\mathbf{Cons\enspace 3}$.
>
> The recent work [4,Theorem 4.1] established a fundamental lower bound of order $ \Omega\left( \left( \frac{ dL ( f(x^1)-f^* ) ||\sigma||_1^2 }{T} \right)^{1/4} \right) $ for SGD when measuring gradients by $\ell_1$-norm. Notably, when $||\sigma||_1 \approx \sqrt{d} ||\sigma||_2 = \sqrt{d} \sigma_s$ (as typically expected), this lower bound precisely aligns with our convergence rate in Eq. (3). We further conjecture that this lower bound applies more broadly to general first-order stochastic optimization algorithms under $\ell_1$-norm gradient measurement. This would imply that our derived convergence rate is nearly tight.
>
> [4]. R. Jiang, et al. Convergence analysis of adaptive gradient methods under refined smoothness and noise assumptions. COLT 2025.
>
> $\mathbf{Response\enspace to\enspace Questions}$:
>
> $\mathbf{Question\enspace 1}$.
>
> To the best of our knowledge, existing convergence analyses for adaptive gradient methods can be broadly categorized into two frameworks. The first framework (e.g., [5-7] for Adam and [8] for RMSProp) requires $\beta_2=1-O(1/T)$ but imposes no constraints on $\beta_1$. While this approach achieves weak $\varepsilon$-dependence (typically $\log\frac{1}{\varepsilon}$), it often results in complex proofs with stronger dependence on terms $L$, $\sigma_s$, $f(x^1)-f^*$, and especially the dimension $d$ (only for Adam, excluding the simpler AdaGrad and RMSProp). The second framework (e.g., [9]) instead requires $\beta_1=1-O(1/\sqrt{T})$ without restricting $\beta_2$, which generally yields simpler proofs and tighter convergence rates but introduces explicit $\varepsilon$-dependence (typically $1/\varepsilon^a$ for some $a$). The second proof framework attains these advantages because the proof is highly dependent on $\varepsilon$.
>
> Our work adopts the second framework while innovatively addressing its explicit $\varepsilon$-dependence in the convergence rate through: (i) careful bounding of the second moment estimate via Eq. (12) (line 153), and (ii) strategic setting of $\varepsilon=\frac{\sigma_s^2}{d}$ (such that $K ||\sigma||_1 \leq Kd \sqrt{\varepsilon}$) to eliminate the explicit dependence on $\varepsilon$ in term (k) (line 163). This approach effectively achieves the tight convergence rate without explicit dependence on $\varepsilon$.
>
> [5]. A. Defossez, et al. A simple convergence proof of Adam and AdaGrad. TMLR 2022.
>
> [6]. Y. Hong, et al. On convergence of Adam for stochastic optimization under relaxed assumptions. NeurIPS 2024.
>
> [7]. B. Wang, et al. Closing the gap between the upper bound and the lower bound of Adam’s iteration complexity. NeurIPS 2023.
>
> [8]. S. Xie, et al. Adam exploits $\ell_{\infty}$-geometry of loss landscape via coordinate-wise adaptivity. ICLR 2025.
>
> [9]. H. Li, et al. Convergence of Adam under relaxed assumptions. NeurIPS 2023.
>
> $\mathbf{Question\enspace 2}$.
>
> The reviewer is correct. From the KKT conditions in Eq. (2), we can characterize the convergence by
>
> $||x||_{\infty}\leq \frac{1}{\lambda}$ and $ < \lambda x , \nabla f(x) > + || \nabla f(x) ||_1 \leq \epsilon$.
>
> When the trajectory's infinity norm is bounded as $||x||_{\infty}\leq \frac{c}{\lambda}$ for some $c<1$, we derive the inequality:
>
> $< \lambda x , \nabla f(x) > + || \nabla f(x) ||_1 \geq (1-c) || \nabla f(x) ||_1$
>
> This inequality establishes that convergence in the gradient's $\ell_1$-norm directly implies convergence to KKT points.

---

> > ### Comment · Reviewer_FJAz · 2025-08-06
> >
> > Thanks for authors' detailed explanation of my questions. My questions are mostly settled. I will raise my scores accordingly.

---

> > > ### Author Response · Authors · 2025-08-06
> > >
> > > We sincerely thank you to review our rebuttal and are truly grateful for increasing the score. Thank you again most kindly.

---

### Decision · Program_Chairs · 2025-09-17

**Decision:**

Accept (poster)

**Comment:**

The paper establishes the first convergence rate, $O(\frac{\sqrt{d}}{T^{1/4}})$, for the AdamW optimizer under the practical condition of a constant, non-vanishing weight decay.

**Strengths**: This paper provides a novel theoretical result for a widely-used optimizer, filling a significant gap in the literature. The proofs are rigorous, and the techniques for handling the epsilon term in the denominator are innovative. The paper is well-written and clearly positions its contribution.

**Weaknesses**: The hyperparameter settings required by the theory are more restrictive than those used in practice. Also the dependency on epsilon is polynomial and sensitive. It requires knowing the gradient noise covariance to set the epsilon for the proved rate. The analysis also omits the bias-correction terms used in standard implementations.

**Reason for Acceptance**:
This paper provides a foundational non-convex convergence analysis for a key optimizer in modern machine learning, AdamW. The reviewers all agree on the paper's novelty and praise the clarity of presentation. It is a clear accept.

**Rebuttal Summary**:
Reviewers raised concerns about the practicality of convergence result due to its narrow range of weight decay coefficient and the choice of the $\ell_1$ norm. The authors effectively addressed these with new experiments on large models and strong theoretical justifications. This productive discussion resolved the main concerns and led two reviewers to raise their scores, solidifying the consensus for acceptance.